

**Identification of erosion hotspots and scale-dependent runoff controls on**
**sediment transport in an agricultural catchment**
Christopher Thoma[1,2], Borbala Szeles[1,2], Miriam Bertola[1,2], Elmar M. Schmaltz[3], Carmen
Krammer[3], Peter Strauss[3], Günter Blöschl[1,2]
[1] Institute of Hydraulic Engineering and Water Resources Management, Vienna University of
Technology, Vienna, Austria
[2] Centre for Water Resource Systems, Vienna University of Technology, Vienna, Austria
[3] Institute for Land and Water Management Research, Federal Agency for Water
Management, Petzenkirchen, Austria
**Corresponding author:** Christopher Thoma
**E-mail address:** thoma@hydro.tuwien.ac.at
**Abstract:**
Understanding how agricultural land management influences sediment transport is crucial for
identifying critical source areas (CSAs) and improving erosion mitigation strategies. While
numerous studies focus on in-stream sediment concentrations, fewer investigate overland
flow on the hillslopes. We monitored streamflow and sediment fluxes at an overland flow
station (E2) and an in-stream station (MW) across 55 runoff events (2012–2022) in the
Hydrological Open Air Laboratory (HOAL), Austria. The catchment was segmented into four
distinct areas (A, B, GW9, C) based on topography, hydrological connectivity, and proximity to
the stream, allowing a spatially explicit assessment of erosion hotspots. Temporal patterns of
sediment transport were analysed to infer spatial variability, and differences in sediment
transport dynamics among areas were quantified using Kruskal-Wallis tests and effect size
analysis. Results suggest that at E2 (hillslope scale), non-erosive cultivation significantly
reduced peak turbidity (~9.5 times) and sediment load (~3.8 times) in flat agricultural areas
(7.2% slope, <500 m from the stream) but had no measurable effect in steep (10–12% slope)
or distant (>1000 m) agricultural areas. Across all field types, conversion to non-erosive
cultivation did not affect peak flow. At MW (catchment scale), compared to E2, peak turbidity
at MW decreased (~5.4–7.7 times) due to dilution from subsurface flow contributions, while
peak flow increased (~2.8–11 times) due to additional inputs from wetlands, springs, and
subsurface flows. Sediment load at MW was ~2.4–5.4 times higher than at E2, likely due to
unmonitored diffuse overland flow and sediment inputs from tile drainages. Our findings
indicate that non-erosive cultivation alone in steep terrains or distant agricultural areas is
insufficient to effectively mitigate sediment transport. Effective sediment management in
agricultural catchments requires spatially targeted erosion control strategies that account for
topography, hydrological connectivity, and field proximity to streams.





## 1. Introduction


The loss of topsoil due to erosion leads to decreased soil fertility, reduced organic matter
content, and diminished water retention capacity, ultimately impairing agricultural
productivity and food security (Borrelli et al., 2020; FAO, 2020). Beyond agricultural losses,
sediment transport into water bodies introduces nutrient imbalances, particularly phosphorus
and nitrogen, which drive eutrophication and harmful algal blooms, degrading freshwater
resources (Akinnawo, 2023).
Agricultural catchments are particularly vulnerable to erosion during intense rainfall events,
where overland flow mobilizes soil from cultivated fields (Firoozi and Firoozi, 2024). The
magnitude of sediment transport depends on natural drivers—including precipitation
intensity, soil moisture, and slope (Nearing et al., 2017; Kirkby et al., 2000; Fryirs, 2013)—as
well as anthropogenic influences, such as tillage intensity and cropping systems (Boardman
and Poesen, 2006; Lal, 2015). While natural erosion drivers cannot be controlled, targeted
land management strategies can significantly reduce sediment transport to streams (Tomer
and Schilling, 2009; Doody et al., 2017). However, the effectiveness of these strategies varies
across landscape position, topography, and hydrological connectivity.
The Critical Source Area (CSA) concept suggests that a small fraction of the landscape (~20%)
contributes the majority (~80%) of sediment yield (Pionke et al., 2000). CSAs are typically
characterized by steep slopes, direct stream connectivity, and rapid runoff responses, making
them dominant sediment sources (Tomer and Schilling, 2009; Doody et al., 2017). Overland
flow is often a dominant mechanism for transporting eroded soil from CSAs into stream
networks, where sediment load dynamics can be further shaped by hydrological processes
such as dilution, deposition, tile drainage, and subsurface contributions (Pastén-Zapata et al.,
2014; King et al., 2015). However, the role of these hydrological modifications in shaping in-
stream sediment transport remains insufficiently understood, particularly in agricultural
settings where both surface and subsurface pathways interact to redistribute sediment.
Understanding these linkages is essential for accurately predicting in-stream sediment
concentrations and assessing the effectiveness of erosion mitigation strategies. However,
effective implementation of conservation strategies relies on correctly identifying CSAs first -
without precise spatial targeting, erosion control efforts may be ineffective or misallocated
(Giri et al., 2016). Thus, developing robust, data-driven approaches to pinpoint high-risk
sediment sources is essential for implementing erosion mitigation strategies.
Soil conservation practices—such as conservation tillage, cover crops, and contour farming—
can effectively reduce sediment transport in areas where overland flow velocities are low,
allowing deposition before reaching streams (Gumiere et al., 2011; Fryirs, 2013). However, in
steep terrain (≥10% slopes), high flow velocities and short retention times reduce their
effectiveness, allowing continued sediment transport despite mitigation efforts (Boardman
and Poesen, 2006). Fields with direct hydrological connections to streams contribute
significantly more sediment than isolated fields, where material can settle before entering
waterways (Li et al., 2021; U.S. Environmental Protection Agency, 2015).
While conservation practices at the plot or field scale have been widely studied (Maetens et
al., 2012; Her et al., 2015), CSA identification remains rarely validated through long-term, high-





resolution sediment monitoring. As a result, the true effectiveness of erosion mitigation
strategies remains poorly constrained at the catchment scale, and the influence of overland
flow and sediment redistribution on conservation outcomes is still unclear (Doody et al., 2017,
Van Oost et al., 2007; Verstraeten et al., 2002). This gap in understanding how sediment is
transported from fields to stream networks limits the ability to assess conservation
effectiveness at early transport stages. Long-term studies assessing the interaction between
surface and subsurface transport in sediment connectivity remain scarce, despite advances in
erosion monitoring.
Peak flow, turbidity, and sediment load are influenced by hydrological processes and sediment
dynamics as overland flow transitions to in-stream conditions. At the in-stream scale,
subsurface flow contributions dilute sediment-water suspension, reducing turbidity and
sediment concentrations, particularly in catchments with groundwater discharge or wetland
buffering (Pastén-Zapata et al., 2014; King et al., 2015; Exner-Krittidge et al., 2016). However,
peak flow often increases due to wetland, spring, and subsurface flow contributions,
amplifying baseflow into stream networks (Blann et al., 2009).
Tile drainage systems further complicate sediment transport dynamics by bypassing
traditional sediment retention mechanisms, allowing fine sediments to enter streams directly
(Gentry et al., 2007; Rittenburg et al., 2015). Additionally, unmonitored diffuse overland flow
paths increase uncertainty in linking hillslope-scale erosion control to in-stream sediment
loads (Sharpley et al., 2009; Doody et al., 2017).
To address these knowledge gaps, this study leverages a decade-long, high-resolution dataset
from the Hydrological Open Air Laboratory (HOAL) in Austria (Blöschl et al., 2016), providing a
unique opportunity to assess conservation effectiveness at both field and catchment scales.
In particular, we examine how sediment transport processes operate across different spatial
scales, tracking changes from overland flow contributions at the hillslope scale to in-stream
sediment transport at the catchment scale. We examine how land management, topography,
and hydrological pathways influence sediment fluxes across multiple agricultural fields
differing in slope gradients, location within the catchment, and connectivity to the stream
network. A key focus is on identifying CSAs where non-erosive cultivation could significantly
reduce sediment export to the stream. Thus, we aim to address the following key research
questions:
• Which field characteristics determine the effectiveness of non-erosive cultivation
practices in reducing turbidity, sediment load, and overland flow?
• To what extent can a shift in cultivation practices mitigate sediment yield and water
fluxes?
• How do peak flow, turbidity, and sediment load change across spatial scales, from
overland flow inputs to in-stream monitoring stations, and what scale-dependent
hydrological and sediment transport processes drive these changes?




## 2. Study Area and Data

### 2.1 Study Area

The HOAL is a 66 ha agricultural catchment located in Petzenkirchen, Lower Austria (Figure 1). It is situated in a humid temperate climate, with a mean annual temperature of 9.5°C and mean annual precipitation of 823 mm (1990–2014), peaking during summer convective storms. These high-intensity precipitation events frequently trigger overland flow, which serves as a key driver of sediment mobilization and transport within the catchment (Blöschl et al., 2016).

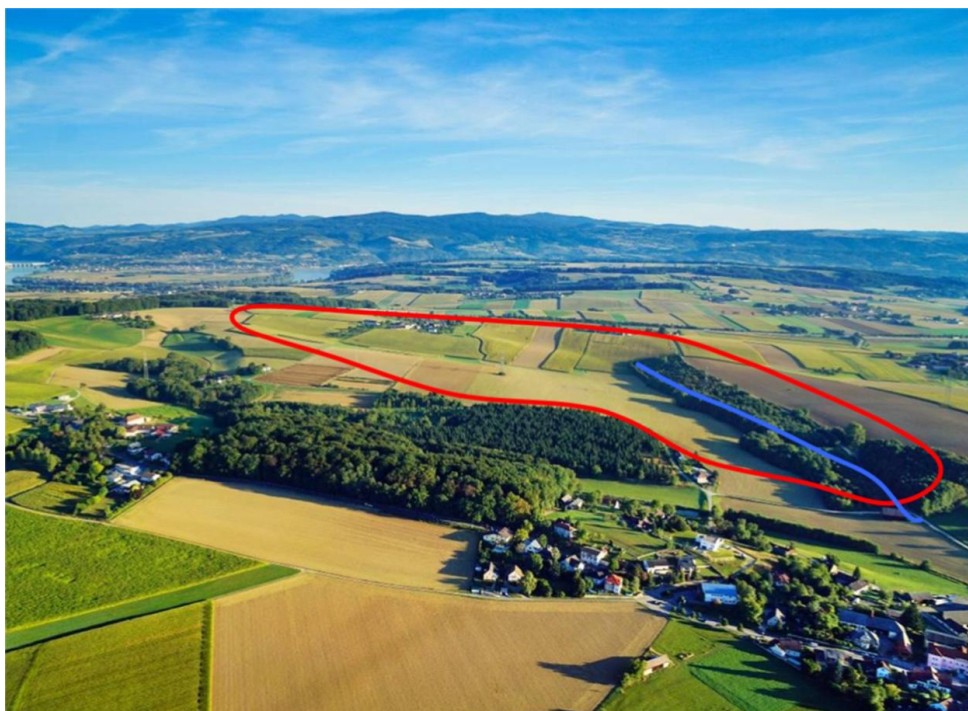

*Figure 1: Aerial photograph of the 66 ha Hydrological Open Air Laboratory (HOAL) in lower Austria. Red line indicates the topographic catchment boundary, blue line indicates the main stream of the catchment. (Photo: BAW/Alexander Eder)*

Elevation ranges from 268 to 323 m, with an average slope of 8%, creating spatial variability in overland flow generation and sediment connectivity. The underlying geology consists of Tertiary fine sediments of the Molasse zone, underlain by fractured siltstone, which influences subsurface drainage and sediment retention capacity. The dominant soil types—Cambisols (57%), Planosols (21%), Kolluvisols (16%), and Gleysols (6%)—exhibit moderate to poor infiltration capacities, with clay contents between 20–30%. Notably, Planosols and Gleysols contain low-permeability clay layers, leading to frequent waterlogging that enhances overland flow and erosion risk, particularly in intensively cultivated areas (Blöschl et al., 2016).

The Seitengraben, the main stream in the catchment, spans 620 m, with its outlet monitored at station MW. Water and sediment inputs originate from multiple surface and subsurface sources, which shape the sediment transport pathways.




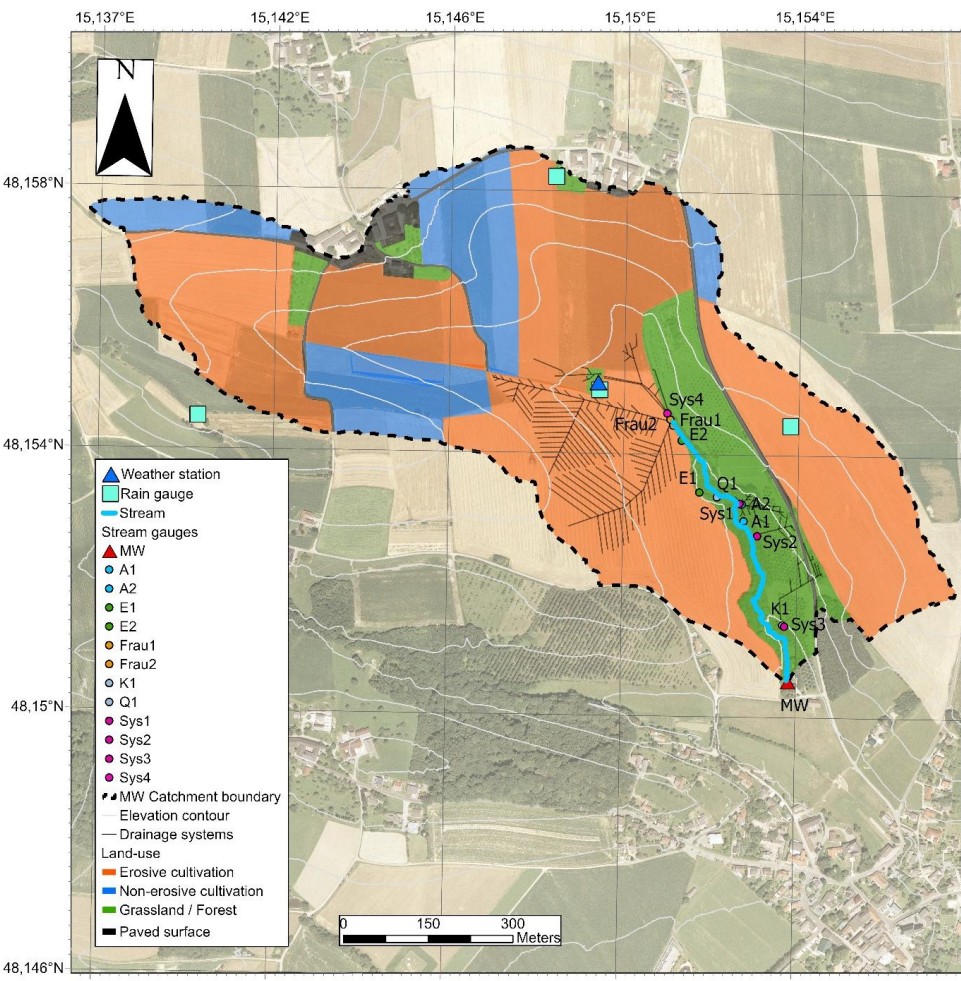


*Figure 2: Overview map of the HOAL in Petzenkirchen, Austria. The map shows the spatial distribution of land-use classes*
*within the catchment for the year 2015 as an example. Additionally, it highlights the drainage systems, runoff measurement*
*stations, catchment boundary, the Seitengrabenbach stream, and the location of the rain gauges. (Credit: Credit: base map*
*layers from BAW-IKT; modifications and annotations by the authors)*
The main hydrological and sediment transport pathways include:
• Erosion Gullies (E1, E2) (Figure 3a and b): Overland flow is the dominant driver of
sediment transport during intense rainfall events, mobilizing eroded material from
agricultural fields and channelled through the erosion gullies, which serve as highly
connected sediment pathways to the stream.
• Tile Drainage Systems (Sys1, Sys2, Sys3, Sys4, Frau1, Frau2) (Figure 3c and d): Installed
between the 1930s and 1950s to mitigate waterlogging, these systems bypass natural
sediment retention mechanisms, transporting fine-grained sediments directly to the
stream. Covering 15% of the catchment they facilitate the direct transport of eroded



material into the stream network (Blöschl et al., 2016). Among these, Frau 1 and Frau
2 are particularly relevant, as they demonstrably contribute to unmonitored sediment
fluxes. Unlike overland flow, which primarily mobilizes coarser sediment fractions, tile
drains transport fine sediment particles, which can accumulate within the drainage
pipes. During high-flow conditions, such as intense precipitation events, periods of
high antecedent soil moisture, or when preferential flow occurs through macropores,
these sediments can be flushed out, producing sudden pulses of sediment delivery to
the stream.
• Springs and Wetlands (Q1, K1, A1, A2) (Figure 3e and f): Perennial and intermittent
spring discharges (Q1, K1) contribute to baseflow, generally diluting turbidity, though
colloidal sediment transport may occur under certain conditions. A1 and A2 function
as wetlands that periodically dry out, influencing local water retention and sediment
deposition dynamics rather than providing constant baseflow contribution.
In addition to monitored inflow pathways, several unmonitored diffuse inflow pathways
contribute to sediment transport in the catchment. During large runoff events, excess water
and sediment can enter the stream through unmonitored paths between monitored stations
E1 and E2 (Figure 3g), diffuse overland flow pathways that are not fully captured by monitoring
infrastructure, and subsurface tile drainage systems at Frau 1 and Frau 2 (Figure 3h). These
pathways may lead to undetected sediment fluxes to the stream.





The land-use within the catchment is predominantly agricultural (Figure 2), with 87% of the
area usually cultivated with winter wheat, maize, barley, and oilseed rape. The remaining land
comprises forest (6%), grassland (5%), and paved surfaces (2%).
Understanding how these sediment pathways interact under different land management
strategies is essential for identifying CSAs and evaluating their contributions to in-stream
sediment fluxes.

### 2.2 Instrumentation and data availability

Precipitation was available at one-minute temporal resolution using four OTT Pluvio² weighing
rain gauges, each with a 400 cm² collecting area. The gauges were strategically installed across
the catchment to capture spatial precipitation variability (Figure 2).
Runoff was monitored at the overland flow station E2, the catchment outlet MW, and
additional tributary stations using calibrated H-flumes equipped with Druck PTX1830 pressure
transducers, providing one-minute temporal resolution. These stations capture diverse runoff
generation and sediment transport pathways, including overland flow (E2), tile drainage
(Sys1–Sys4, Frau1, Frau2), wetlands (A1, A2), and springs (K1, Q1).
To complement discharge monitoring, turbidity measurements were conducted at MW, E1,
and E2 using WTW ViSolid optical sensors (Xylem Analytics Germany, 2018). These sensors
determine suspended sediment concentrations (mg/l) via scattered light and backscattering
techniques. To maintain accuracy, the sensors were equipped with an ultrasonic cleaning
system to prevent biofouling and contamination by sediment. The raw turbidity data was
calibrated using laboratory analyses of total suspended solids (TSS) from collected water
samples.
A comprehensive water sampling program was conducted from 2012 to 2022 to analyze
sediment dynamics. Manual samples were collected monthly at all discharge stations when
flow was present. For event-based sampling, ISCO 6712 automatic samplers (Teledyne ISCO,
2019) were deployed using station-specific thresholds (Blöschl et al., 2016; Szeles et al., 2024).
When water levels in the H-flumes exceeded pre-defined thresholds—adjusted based on
baseflow conditions—sequential one-liter samples were collected at intervals ranging from 15
minutes to 2 hours, depending on event duration. Sampling continued until all 24 bottles were
filled or water levels dropped below the threshold. Following each event, the samples were
refrigerated and replaced within three days to ensure data integrity. A comprehensive
description of the instrumentation and data availability is provided by Blöschl et al. (2016).



In addition to hydrological monitoring, detailed land management information was provided
by farmers from 2012 to 2022. This dataset included crop rotation, fertilization regimes, along
with records of tillage, sowing, and harvesting schedules for all parcels within the catchment.

## 222   3. Methods

### 223   3.1 Data preparation

Given the rapid runoff response of the catchment to precipitation (Chen et al., 2020),
discharge data at MW and E2 was analysed at a 5-minute resolution to improve temporal
accuracy. Rainfall measurements from the four OTT Pluvio² gauges were averaged, as
variations between them were minor. The original one-minute precipitation data was
aggregated into 5-minute intervals by summing the recorded values.
To quantify the rainfall erosive potential, the erosivity index $EI_{30}$ was calculated. $EI_{30}$
represents the product of the total kinetic energy of a rainfall event and its maximum 30-
minute rainfall intensity (Wischmeier and Smith, 1978). The calculation follows the equation
232 (1):

$$EI_{30} = \sum_{i=1}^{m} E_i \cdot I_{30,i} \qquad (1)$$

where $EI_{30}$ is the rainfall erosivity factor (MJ mm ha$^{-1}$ h$^{-1}$), $E_i$ is the total kinetic energy per
rainfall event (MJ m$^{-2}$), $I_{30}$ is the maximum 30-minute rainfall intensity within the event (mm
h$^{-1}$), and $m$ represents the number of rainfall events.
Sediment load was estimated by multiplying flow data from MW and E2 with turbidity
measurements.
In terms of agricultural data, the main crops present in the study catchment between 2012
and 2022 included bare soil, soybean, potato, winter wheat, winter barley, maize, and cover
crops such as clover grass and Lucerne. Following the classification outlined in the Austrian
Agricultural Environmental Programme (ÖPUL) and Soil Erosion Evaluation Report (BML,
2021), these crops were categorized into erosive and non-erosive types. Erosive crops
included bare soil, soybean, potato, and maize, while non-erosive crops comprised winter
wheat, winter barley, and cover crops. Crop statistics of arable land for the study period are
summarized in Table 1.
*Table 1: Crop statistics of arable land between 2012 and 2022. The classification into erosive and non–erosive crop types*
*was done according to the Austrian Agricultural Environmental Programme (ÖPUL) and Soil Erosion Evaluation Report (BML,*
*2021).*

| Year | Erosive Cultivation | | Non-erosive Cultivation | |
|---|---|---|---|---|
|  | Area (ha) | Area (%) | Area (ha) | Area (%) |
| **2012** | 1.7–53.3 | 3.0–92.8 | 4.2–55.7 | 7.23–97.0 |
| **2014** | 13.4–52.1 | 23.4–90.8 | 5.3–44.0 | 9.2–76.6 |
| **2015** | 13.4–55.9 | 23.4–90.8 | 1.5–44.0 | 2.6–76.6 |
| **2016** | 1.1–56.1 | 1.9–97.7 | 1.3–56.3 | 2.3–98.1 |
| **2017** | 1.1–55.2 | 1.9–96.1 | 2.2–56.3 | 3.9–98.1 |
| **2018** | 10.6–55.3 | 18.4–96.3 | 2.1–46.9 | 3.7–81.6 |
| **2019** | 7.7–56.5 | 13.4–98.4 | 0.9–49.7 | 1.6–86.6 |
| **2020** | 7.7–52.6 | 13.4–91.7 | 4.8–49.7 | 8.3–86.6 |



| | | | | |
|---|---|---|---|---|
| **2021** | 6.3–51.5 | 11.0–89.7 | 5.9–51.0 | 10.3–89.0 |
| **2022** | 3.4–43.6 | 5.9–76.0 | 13.8–54.0 | 24.0–94.1 |


### 3.2 Event separation

At the E2 station, hydrological events were measured exclusively during periods of overland
flow generation triggered by precipitation events, aligning with findings from previous studies.
To identify periods of direct flow at the MW station in the time series, we applied an
automated recursive digital filter (Nathan and McMahon, 1990; Arnold et al., 1995). This
approach is particularly well-suited for our study catchment, given its demonstrated
effectiveness in separating quick response flows from baseflows (Eder et al., 2010). Prior to
applying the digital filter, noisy discharge data were smoothed using a moving average filter
with a 5-minute window size.
The recursive digital filter is governed by the following equation (2):

$$q_t = \beta \cdot q_{t-1} + \frac{(1+\beta)}{2} \cdot (Q_t - Q_{t-1}) \tag{2}$$

where $q_t$ is the filtered quick response (event water) at time step t, $Q_t$ is the total streamflow,
and $\beta$ is the filter parameter. Following the recommendations of Nathan and McMahon (1990)
and Arnold et al. (1995), the value of $\beta$ was set to 0.95, which was verified as appropriate for
the catchment through visual data inspection.
In defining hydrological events for our study, we followed a similar approach to Eder et al.
(2010). An event was considered to begin when streamflow increased above baseflow and
ended when only baseflow contributed to discharge, even if baseflow sediment
concentrations remained elevated. To ensure that small fluctuations did not lead to an
excessive number of minor events, additional thresholds were applied: the maximum
discharge of an event had to reach at least 5 ls$^{-1}$, the increase in discharge from its preceding
value had to exceed 2 ls$^{-1}$, and the peak turbidity during the event had to surpass 100 mgl$^{-1}$.
These criteria effectively minimized the inclusion of low-flow events with negligible suspended
sediment transport.

### 3.3 Event selection

Between 2012 and 2022, we identified 255 events at MW and 85 at E2. However, while MW
events were consistently recorded, frequent sensor failures at E2 (e.g., turbidity sensors, flow
sensors, ISCO samplers) reduced the number of fully recorded events to 22. Most missing data
affected the rising or falling limb of hydrographs and turbidigraphs, limiting the dataset for E2.
To address scale effects between overland flow at E2 and in-stream dynamics at MW, we
matched corresponding E2 events with MW events whenever possible. Given the strong
correlations observed between peak values and total event values at both stations (Figure 4),
peak values were used as proxies to expand the dataset from 22 to 55 events at E2. This
approach facilitated a more comprehensive cross-scale comparison while mitigating the
impact of missing data at E2.




Statistical validation confirmed the reliability of peak values as event proxies. At E2, peak flow
and total event flow were strongly correlated (r = 0.77, Figure 4a), while peak sediment load
(kg/5 min) and total event sediment load (kg) exhibited an even stronger correlation (r = 0.93,
Figure 4b). At MW, these relationships were even more pronounced (r = 0.84 and r = 0.95,
Figures 4c and 4d, respectively). These findings align with previous research (e.g., Skålevåg et
al., 2024), demonstrating that peak flow and peak sediment load reliably characterize event-
scale sediment transport in highly responsive catchments.
We excluded 2013 from the analysis, as stream maintenance and excavator work at MW's
right bank artificially increased sediment loads, biasing the dataset. While E2 measurements
remained unaffected, the lack of reliable MW data precluded its use for that year.

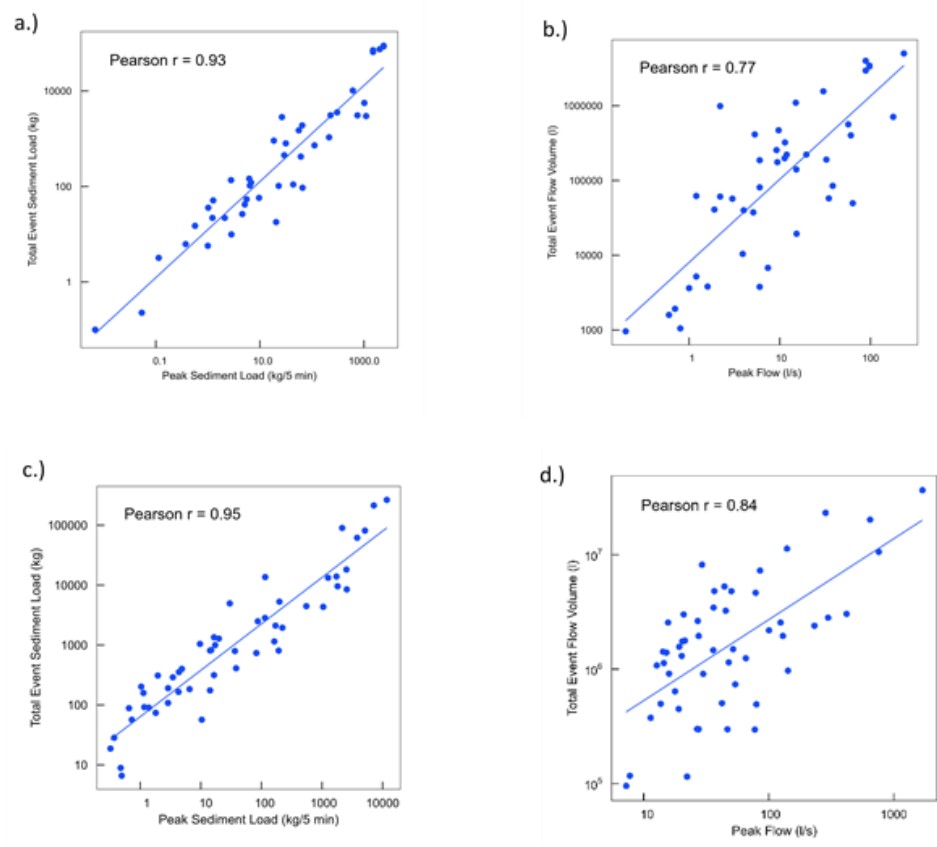


*Figure 4: Relationship between (a) peak sediment load (kg/5 min) and total event sediment load (kg) at site E2, (b) peak flow*
*(l/s) and total event flow volume (l) at site E2, (c) peak sediment load (kg/5 min) and total event sediment load (kg) at site*
*MW, (d) peak flow (l/s) vs. total event flow volume (l) at site MW, with data points corresponding to individual events. Both*
*axes are displayed on a logarithmic scale, and the blue line indicates the linear regression fit. The Pearson correlation*
*coefficient (r) is displayed in each panel.*
## 3.4 Catchment Segmentation
In the HOAL, sediment transport is highly event-driven and influenced by a complex
interaction of hydrological connectivity, land management, and topography. Observations





from field visits, photographic evidence, and previous hydrological modelling studies (Strauss
et al., 2007) confirm that sediment transport rates are not only influenced by local field
conditions (e.g., topography, cultivation) but also by the spatial arrangement of fields within
the catchment.
Two primary overland flow pathways transport sediment from agricultural fields to the
stream, merging in the flat convergence zone in area C before reaching the E2 station (Figure
5). The steep slope of flowpath 2 promotes higher flow velocities and greater sediment
transport capacity than flowpath 1. Prior to reaching overland flow station E2, flowpath 2
traverses area GW9— a steeply inclined area that may act either as a sediment source or a
sink, depending on its vegetative cover, which alternates between erosive and non-erosive
cultivation. Similarly, flowpath 1 intersects area C, a comparatively flatter area where
sediment may be deposited or mobilized further, depending primarily on cultivation practices.
This segmentation of the HOAL catchment is based on the presence of these two distinct
flowpaths, each contributing overland flow and sediment to the same outlet (E2) but following
different trajectories through the catchment. Crucially, the final area traversed by flowpath 1
is flat and long, while that of flowpath 2 is steep and shorter. This configuration provides an
ideal natural experiment to assess how cultivation type (erosive vs. non-erosive) interacts with
terrain steepness to influence sediment and water retention in the agricultural landscape. It
allows for the isolation of topographic and cultivation type effects on sediment yield along key
hydrological flowpaths.
To systematically evaluate the spatial influence of different areas with distinct characteristics
on sediment yield at E2, we segmented the HOAL catchment into four distinct zones (Figure 5
and Table 2). This classification allows us to quantify how slope, distance, and connectivity
influence sediment yield, facilitating a spatially explicit assessment of erosion hotspots.
• Area A: This zone represents the source of flowpath 1, consisting of agricultural fields
within a larger sub-catchment with moderate slopes (weighted mean: 9.7%) and a
greater distance from the stream (weighted mean: 1112 m).
• Area B: The source of flowpath 2, characterized by agricultural fields with steeper
slopes (weighted mean: 10.3%), and a medium distance from the stream (weighted
mean: 840 m).
• Area GW9: This area represents the final agricultural field before flowpath 2 merges
with flowpath 1 in Area C. It is situated at a moderate distance from the stream
(weighted mean: 763 m) and has a very steep slope (weighted mean: 11.5%).
• Area C: The final transition zone where both flowpath 1 and flowpath 2 converge
before entering the stream. This area is flat (weighted mean: 7.2%) and in immediate
proximity to the stream (weighted mean: 492 m).
By integrating this framework with high-resolution sediment monitoring, we can directly
identify high-risk areas and evaluate their contributions to both overland flow and in-stream
sediment transport. Furthermore, this enables a targeted evaluation of conservation
effectiveness along key flow pathways, particularly in high-connectivity areas. It also lays the
foundation for analyzing scale-dependent sediment transport, addressing our objective of
identifying CSAs while setting the stage for cross-scale sediment flux analysis.



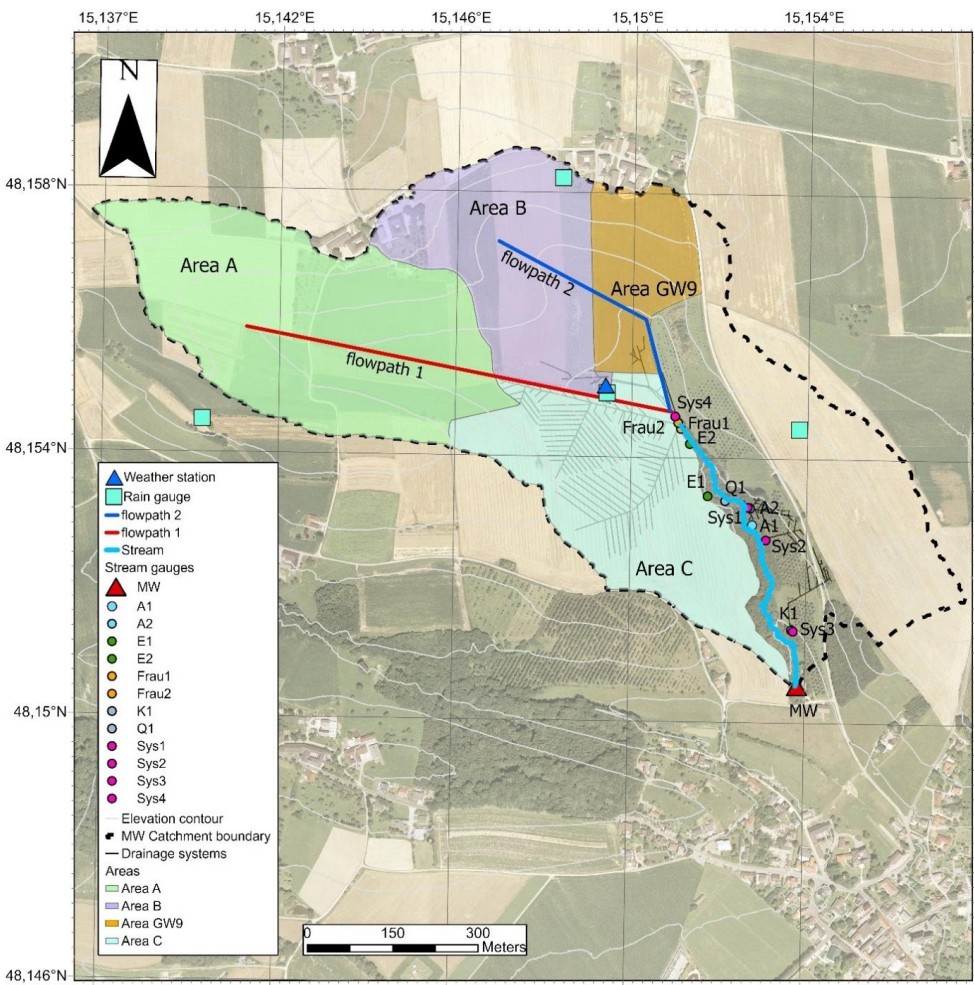

Figure 5: The map illustrates the division of the HOAL catchment into four distinct categories based on topographical and hydrological characteristics: (A) Green – medium steepness, located far from the Seitengrabenbach stream; (B) Violet – high steepness, situated medium distant to the stream; (GW9) Orange – high steepness, situated close to the stream; and (C) Violet – flat terrain near the stream. Additionally, the map highlights the two main overland flow paths. (Credit: Credit: base map layers from BAW-IKT; modifications and annotations by the authors)

Table 2: Mean weighted slope (%) and flow length (m) for the analyzed areas (A, B, GW9, and C). The slope represents the weighted mean terrain slope, while the flow length indicates the weighted mean distance water travels over the surface before reaching the overland flow measuring station E2.

|                 | Area A | Area B | Area GW9 | Area C |
|-----------------|--------|--------|----------|--------|
| Slope (%)       | 9.7    | 10.3   | 11.5     | 7.2    |
| Flow length (m) | 1112   | 840    | 763      | 492    |

## 3.5 Statistical analysis

To evaluate the influence of cultivation type (erosive vs. non-erosive) on hydrological parameters, we applied the Kruskal-Wallis rank sum test (Kruskal and Wallis, 1952) to analyse differences in peak flow, peak turbidity, and peak sediment load across the study areas A, B,



GW9, and C at the overland flow measurement E2 station during 55 erosive events. The effect
size $\varepsilon^2$ (Tomczak and Tomczak, 2014) was computed to quantify the proportion of variance
explained by cultivation type, with p-values greater than 0.05 indicating statistically non-
significant effect.
To assess hydrological and land cultivation interdependencies, Spearman correlation
coefficients were calculated separately for overland flow (hillslope-scale) and in-stream
measurements (catchment-scale). For each event, peak values of hydrological and sediment-
related variables—such as $EI_{30}$, peak flow, turbidity, and sediment load—were then correlated
with the percentage of area cultivated with erosive crop species within each classified area.
This approach aimed to identify how rainfall intensity and land cultivation practices affect
hydrological responses and sediment transport at different spatial scales. The overland flow
correlation matrix captured direct overland flow effects, while the in-stream correlation
matrix integrated multiple hydrological contributions. This comparison enabled an evaluation
of how land management and hydrological processes influence sediment transport across
spatial scales within the study catchment.

## 4. Results

### 4.1 Cultivation Type Effects on Hydrological Response

#### 4.1.1 Upstream erosion gully

At the E2 station, peak flow, peak turbidity, and peak sediment load exhibited both similarities
and spatial variability across study areas A, B, GW9, and C, reflecting distinct hydrological
conditions and sediment connectivity pathways.
In areas A, B, and GW9, peak turbidity (Figure 6a-c and Table 3) and peak sediment load (kg/5
min) (Figure 6e-g and Table 3) showed no significant differences (p > 0.05) between erosive
and non-erosive cultivation types. For peak sediment load, the median sediment load values
in fields with non-erosive cultivation were generally higher than those in fields with erosive
cultivation. However, the Kruskal-Wallis test confirmed that these differences were not
statistically significant, with small effect sizes in all three areas. Similarly, peak turbidity values
showed no consistent trend between cultivation types, with some areas exhibiting higher
values under non-erosive cultivation and others under erosive practices. These results suggest
that non-erosive cultivation was less effective in reducing peak turbidity and peak sediment
load (kg/5 min) in agricultural areas characterized by steeper slopes and greater distances
from the stream — specifically in Area A (moderate slope ~9.7%, distant ~1110 m), Area B
(steeper slope ~10.3%, medium distance ~840 m), and Area GW9 (very steep slope ~11.5%,
medium distance ~760 m).
In contrast, Area C, characterized by a relatively flat slope (7.2%) and close proximity to the
stream (<500 m), exhibited a pronounced sensitivity of the hydrological response to
cultivation type, with non-erosive practices significantly reducing peak turbidity (Figure 6d)
and sediment load (Figure 6h). Statistical analysis confirmed these reductions as highly
significant, yielding large effect sizes for both parameters (Table 3).
Under erosive cultivation, median peak sediment load was 29.5 kg/5 min, approximately ~9.5
times higher than under non-erosive conditions (2.8 kg/5 min), with a statistically significant





effect (p = 0.0022) and a large effect size ($\varepsilon^2$ = 0.1545), indicating that cultivation type
accounted for ~15.5% of the variation in sediment load. Similarly, peak turbidity reached 22.7
g/L under erosive conditions, about ~3.8 times higher than under non-erosive practices (4.7
g/L), with a highly significant effect (p = 0.0001) and a large effect size ($\varepsilon^2$ = 0.2616), explaining
~26.2% of the variation in peak turbidity.





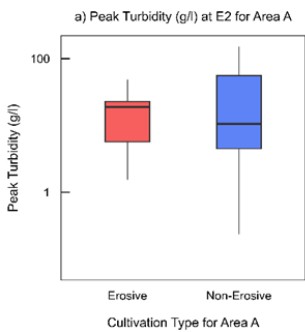

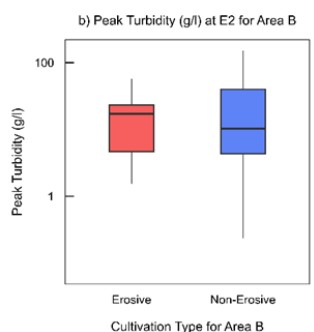

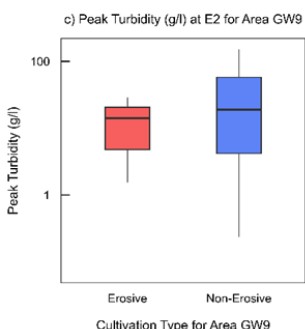

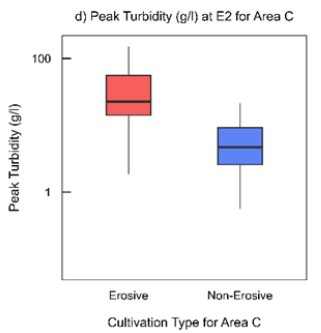

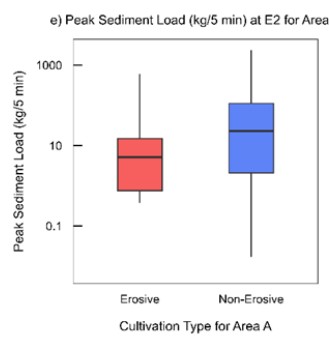

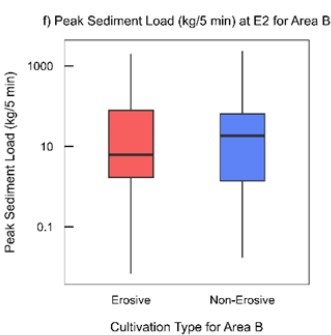

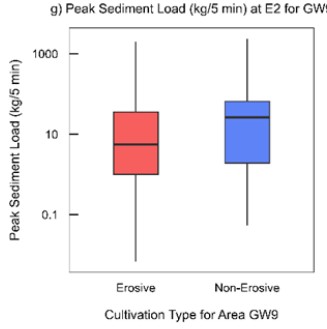

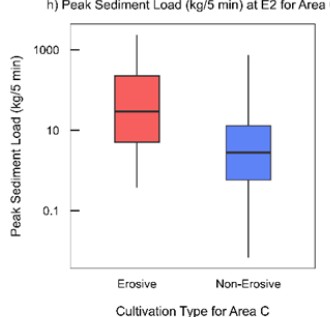




*Figure 6: Boxplots showing the distribution of peak turbidity (g/l), peak sediment load (kg/5 min) at overland flow*
*measurement site E2 for different cultivation types (erosive vs. non-erosive) in areas A, B, GW9, and C. The median and*
*interquartile range are displayed for each category. The y-axis is on a logarithmic scale. Panels correspond to: (a) Peak*
*Turbidity at E2 for Area A, (b) Peak Turbidity at E2 for Area B, (c) Peak Turbidity at E2 for Area GW9, (d) Peak Turbidity at E2*
*for Area C, (e) Peak Sediment Load at E2 for Area A, (f) Peak Sediment Load at E2 for Area B, (g) Peak Sediment Load at E2 for*
*Area GW9, (h) Peak Sediment Load at E2 for Area C.*
In all areas, peak flow (Figure A1 and Table 3) did not differ significantly (p > 0.05) between
erosive and non-erosive cultivation types. While median values varied across locations, the
Kruskal-Wallis test confirmed that these differences were not statistically significant, with
small effect sizes ($\varepsilon^2$) in all areas. These results indicate that cultivation type does not play a
major role in peak flow generation across all agricultural areas with unique hydrological
characteristics, regardless of differences in slope, flow length, or proximity to the stream.
*Table 3: Comparison of peak sediment load (kg/5 min), peak turbidity (g/l), and peak flow (l/s) between erosive and non-*
*erosive cultivation types across different areas at the overland flow measuring station E2. Bold print indicates statistical*
*significance (\*P < 0.05; \*\*P < 0.01; \*\*\*P < 0.001) and a large magnitude of the effect size ($\varepsilon^2$).*

| Hydrological Parameter | Area | Cultivation Type | Peak Value | Kruskal-Wallis χ² (df = 1) | p-value | Effect Size ($\varepsilon^2$) | Magnitude |
|---|---|---|---|---|---|---|---|
| **Sediment Load (kg/5 min)** | A | Erosive | 19.7 | 0.0823 | 0.7743 | -0.017 | Small |
| | | Non-Erosive | 11.5 | | | | |
| | B | Erosive | 16.7 | 0.3164 | 0.5738 | -0.0127 | Small |
| | | Non-Erosive | 14.2 | | | | |
| | GW9 | Erosive | 14.2 | 1.7427 | 0.1868 | 0.0138 | Small |
| | | Non-Erosive | 19.0 | | | | |
| | C | Erosive | 22.7 | 15.126 | **0.0001** | 0.2616 | **Large** |
| | | Non-Erosive | 4.7 | | | | |
| **Turbidity (g/l)** | A | Erosive | 19.7 | 0.0823 | 0.7743 | -0.017 | Small |
| | | Non-Erosive | 11.5 | | | | |
| | B | Erosive | 16.7 | 0.3164 | 0.5738 | -0.0127 | Small |
| | | Non-Erosive | 14.2 | | | | |
| | GW9 | Erosive | 14.2 | 1.7427 | 0.1868 | 0.0138 | Small |
| | | Non-Erosive | 19.0 | | | | |
| | C | Erosive | 22.7 | 15.126 | **0.0001** | 0.2616 | **Large** |
| | | Non-Erosive | 4.67 | | | | |
| **Flow (l/s)** | A | Erosive | 2.2 | 3.9946 | **0.0457** | 0.0555 | Small |
| | | Non-Erosive | 9.2 | | | | |
| | B | Erosive | 4.0 | 0.7341 | 0.3916 | -0.0049 | Small |
| | | Non-Erosive | 6.0 | | | | |
| | GW9 | Erosive | 5.8 | 0.3019 | 0.5827 | -0.0129 | Small |
| | | Non-Erosive | 6.0 | | | | |
| | C | Erosive | 6.0 | 1.1366 | 0.2864 | 0.0025 | Small |
| | | Non-Erosive | 5.9 | | | | |


### 4.1.2 Catchment outlet

The MW station exhibited sediment dynamics similar to those observed at the E2 station, with
no significant differences in peak turbidity (Figure 7a-c), peak sediment load (Figure 7e-g), or



peak flow (Figure A2a-c) for areas A, B, and GW9. While non-erosive cultivation generally
resulted in lower median values for peak turbidity and sediment load, the absence of a
consistent trend suggests that land management practices in steeper (~10–12% slope),
medium-distance (~750–850 m) and moderately steep (~9.6% slope), distant (>1000 m)
agricultural areas have limited influence on sediment transport and turbidity at the catchment
outlet.
In contrast, Area C displayed a significant response to cultivation practices, with erosive
cultivations leading to pronounced increases in both peak turbidity (Figure 7d) and peak
sediment load (Figure 7h), mirroring the trends observed at the E2 station. Events associated
with erosive cultivation resulted in peak sediment loads that were 38 times higher than under
non-erosive conditions (163 kg/5 min vs. 4.3 kg/5 min), with a significant effect (p = 0.0003)
and a large effect size ($\varepsilon^2$ = 0.245), indicating that cultivation type accounts for 24.5% of the
variability in sediment load at the MW station (Table 4). Similarly, peak turbidity was more
than 10 times higher under erosive conditions (5777 g/l) than under non-erosive cultivation
(537 g/l), with a large effect size ($\varepsilon^2$ = 0.2368, p = 0.0003). These results indicate that in flat
agricultural areas (7.2% slope) near the stream (<500 m), non-erosive cultivation significantly
reduces both peak sediment load and peak turbidity at the catchment outlet.



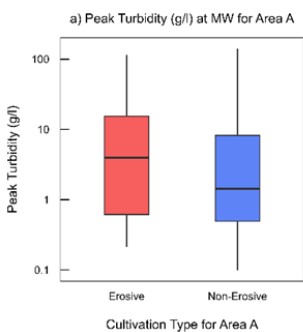

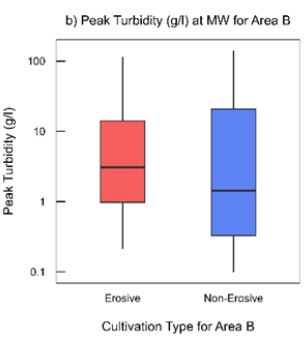

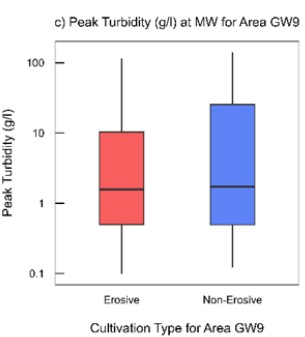

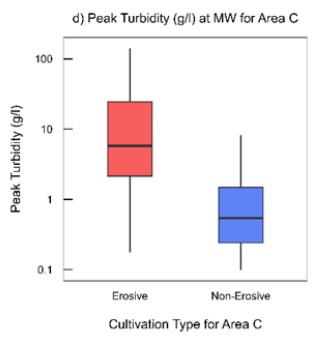

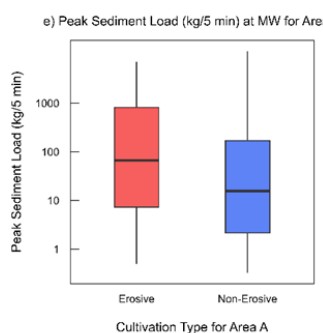

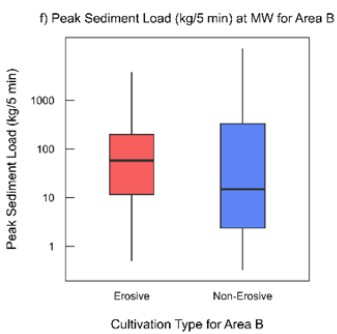

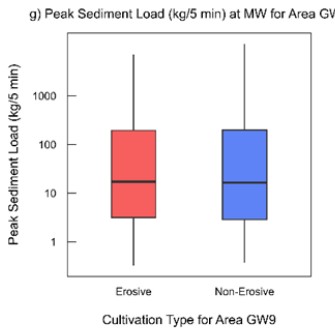

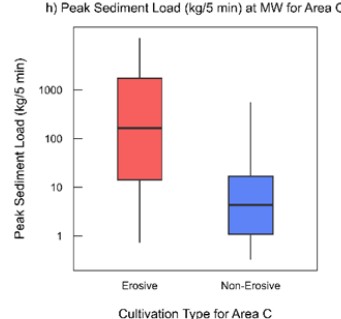






*Figure 7: Boxplots showing the distribution of peak turbidity (g/l) and peak sediment load (kg/5 min) for different cultivation types (erosive vs. non-erosive) at in-stream measurement site MW across areas A, B, GW9, and C. The median and interquartile range are displayed for each category. The y-axis is on a logarithmic scale. Panels correspond to: (a) Peak Turbidity at E2 for Area A, (b) Peak Turbidity at E2 for Area B, (c) Peak Turbidity at E2 for Area GW9, (d) Peak Turbidity at E2 for Area C, (e) Peak Sediment Load at E2 for Area A, (f) Peak Sediment Load at E2 for Area B, (g) Peak Sediment Load at E2 for Area GW9, (h) Peak Sediment Load at E2 for Area C.*

Unlike at E2, where peak flow remained unaffected by cultivation type, the MW station exhibited a statistically significant increase in peak flow under erosive cultivation in Area C (Figure A2d). Median peak flow was approximately three times higher under erosive conditions (65.6 l/s) than under non-erosive practices (22.2 l/s), with a large effect size ($\varepsilon^2 = 0.1571$, p = 0.0029) (Table 4).

*Table 4: Comparison of peak sediment load (kg/5 min), peak turbidity (g/l), and peak flow (l/s) between erosive and non-erosive cultivation types across different areas at the in-stream measuring station MW. Bold print indicates statistical significance (\*P < 0.05; \*\*P < 0.01; \*\*\*P < 0.001) and a large magnitude of the effect size ($\varepsilon^2$).*

| Hydrological Parameter | Area | Cultivation Type | Peak Value | Kruskal-Wallis $\chi^2$ (df = 1) | p-value | Effect Size ($\varepsilon^2$) | Magnitude |
|---|---|---|---|---|---|---|---|
| **Sediment Load (kg/5 min)** | A | Erosive | 38.2 | 0.7026 | 0.4019 | -0.0059 | Small |
| | | Non-Erosive | 16.5 | | | | |
| | B | Erosive | 87.3 | 1.7442 | 0.1866 | 0.0149 | Small |
| | | Non-Erosive | 14.2 | | | | |
| | GW9 | Erosive | 17.2 | 0.0041 | 0.9489 | -0.0199 | Small |
| | | Non-Erosive | 16.5 | | | | |
| | C | Erosive | 163.2 | 13.241 | **0.0003** | 0.245 | **Large** |
| | | Non-Erosive | 4.3 | | | | |
| **Turbidity (g/l)** | A | Erosive | 3977.7 | 0.8821 | 0.3476 | -0.0024 | Small |
| | | Non-Erosive | 1420.5 | | | | |
| | B | Erosive | 3086.3 | 0.4811 | 0.4879 | -0.0104 | Small |
| | | Non-Erosive | 1420.5 | | | | |
| | GW9 | Erosive | 1575.9 | 0.3764 | 0.5395 | -0.0125 | Small |
| | | Non-Erosive | 1721.3 | | | | |
| | C | Erosive | 5777.1 | 12.842 | **0.0003** | 0.2368 | **Large** |
| | | Non-Erosive | 537.0 | | | | |
| **Flow (l/s)** | A | Erosive | 36.0 | 0.11107 | 0.7389 | -0.0178 | Small |
| | | Non-Erosive | 36.7 | | | | |
| | B | Erosive | 65.6 | 2.3986 | 0.1214 | 0.028 | Small |
| | | Non-Erosive | 29.3 | | | | |
| | GW9 | Erosive | 36.7 | 0.57769 | 0.4472 | -0.0084 | Small |
| | | Non-Erosive | 29.3 | | | | |
| | C | Erosive | 65.6 | 8.8537 | **0.0029** | 0.1571 | **Large** |
| | | Non-Erosive | 22.2 | | | | |



## 4.2 Hydrological and Land Cover Interdependencies in Overland Flow and In-Stream Measurements

### 4.2.1 Overland Flow Characteristics

The Spearman correlation analysis (Section 3.4) for overland flow (Figure 8) highlights that rainfall erosivity ($EI_{30}$) is moderately associated with sediment load and turbidity. The moderate positive correlation of $EI_{30}$ with peak sediment load ($\rho = 0.58$) and peak turbidity ($\rho = 0.59$) suggests that more intense rainfall events lead to increased sediment mobilization and higher turbidity levels. The correlation between $EI_{30}$ and peak flow was weaker ($\rho = 0.40$), indicating that while $EI_{30}$ plays a role in overland flow generation, other factors also contribute to shaping flow responses.

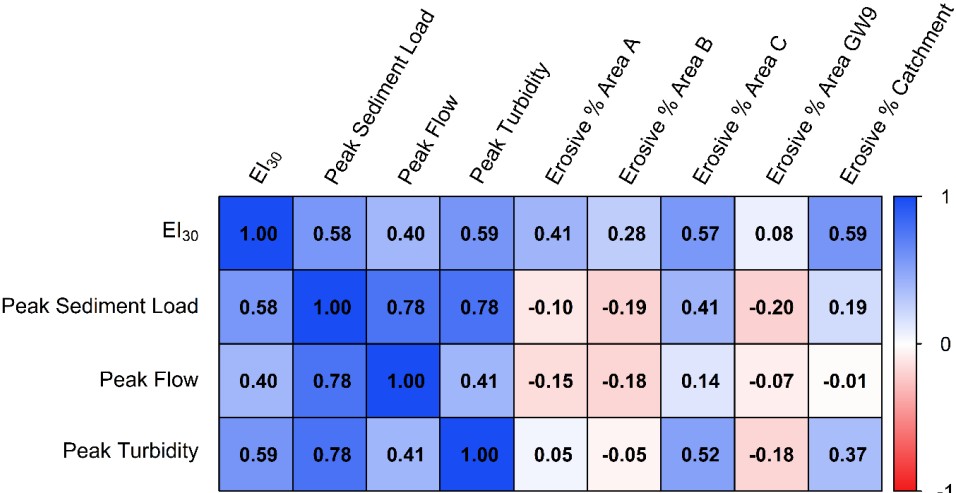

*Figure 8: Spearman correlation matrix for overland flow measurements, showing the relationships between rainfall intensity ($EI_{30}$), hydrological responses (peak sediment load (kg/5 min), peak turbidity (g/l), peak flow (l/s)), and erosive land cover in different catchment areas. Positive correlations are shown in blue, and negative correlations in red, with intensity representing the strength of the correlation.*

Among the hydrological variables, a weak relationship was found between peak turbidity and peak flow ($\rho = 0.41$), suggesting that while flow magnitude influences turbidity, additional factors may contribute to turbidity variations as well.

The influence of erosive land cover on hydrological responses varied across the catchment areas, with spatially heterogeneous relationships between land cover cultivations, and sediment transport dynamics. In Area A, erosive land cover exhibited no relationship with peak turbidity ($\rho = 0.05$). Area B and GW9 displayed even negative correlations between erosive land cover and both peak sediment load (Area B: $\rho = -0.19$, GW: $\rho = -0.20$) and peak turbidity (Area B: $\rho = -0.05$, GW: $\rho = -0.18$). These weak associations suggest that cultivation practices in these areas of the catchment may not have a significant impact on peak turbidity and peak sediment load, as their effect could be obscured by other factors. In contrast, Area C showed a positive correlation between erosive land cover and sediment load ($\rho = 0.41$) and turbidity ($\rho = 0.52$), indicating that in this particular area of the catchment, the type of land cover may exert a stronger influence on sediment transport dynamics.





The relationship between erosive land cover and peak flow was consistently weak and mostly
negative across all areas (Area A: $\rho$ = -0.15, Area B: $\rho$ = -0.18, Area C: $\rho$ = 0.14, GW9: $\rho$ = -0.07).
These results suggest that land cover alone does not strongly explain variations in peak flow.
### 4.2.2 In-Stream Measurement Characteristics
The correlation analysis for in-stream measurements (Figure 9) revealed a stronger influence
of rainfall intensity on hydrological responses compared to overland flow. $EI_{30}$ exhibited higher
correlations with peak sediment load ($\rho$ = 0.79), peak turbidity ($\rho$ = 0.76), and peak flow ($\rho$ =
0.69), indicating that rainfall erosivity exerts a dominant control over hydrological and
sediment transport mechanisms within the catchment and stream network.

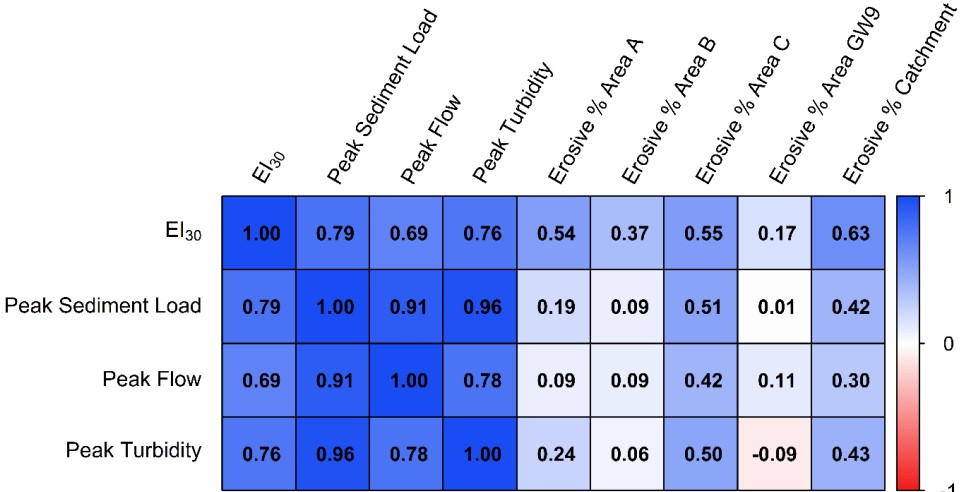


*Figure 9: Spearman correlation matrix for in-stream measurements, showing the relationships between rainfall intensity ($EI_{30}$),*
*hydrological responses (peak sediment load (kg/5 min), peak turbidity (g/l), peak flow (l/s)), and erosive land cover in different*
*catchment areas. Positive correlations are shown in blue, and negative correlations in red, with intensity representing the*
*strength of the correlation.*
The relationships among hydrological parameters were also more pronounced in the in-
stream matrix than in overland flow (Figure 9). Peak flow and peak sediment load ($\rho$ = 0.91)
and peak flow and peak turbidity ($\rho$ = 0.78) exhibited strong associations, suggesting that
sediment and turbidity dynamics in the stream are primarily flow-driven.
The role of land cultivation in in-stream sediment dynamics varied across the catchment.
Correlations between erosive land cover and sediment transport variables were generally
consistent with overland flow, indicating that catchment sediment dynamics is primarily
governed by overland flow and land cover effects rather than diverse in-stream processes. In
Area A and B, erosive land cover exhibited a weak positive relationship with peak turbidity
(Area A: $\rho$ = 0.24, Area B: $\rho$ = 0.05) and peak sediment load (Area A: $\rho$ = 0.19, Area B: $\rho$ = 0.09).
In Area GW9, erosive land cover exhibited a weak negative correlation with peak turbidity ($\rho$
= -0.09) and showed no distinguishable relationship with peak sediment load. These weak
associations suggest that cultivation practices in areas A, B, and GW9 of the catchment do not
have a significant impact on overall peak turbidity and peak sediment load. In contrast, Area
C exhibited the highest correlations between erosive land cover and both peak sediment load



(ρ = 0.51) and peak turbidity (ρ = 0.50) among all areas. While these correlations remain moderate, they may point to a relatively stronger association between land cover and sediment dynamics in this part of the catchment.

The correlation between erosive land cover and peak flow remained consistently weak or negative across all areas (Area A: ρ = 0.09, Area B: ρ = 0.09, Area C: ρ = 0.42, GW9: ρ = 0.11), further supporting the dominant role of rainfall erosivity (ρ = 0.69) in driving streamflow generation rather than direct land cover effects. These results align with findings from the overland flow analysis, reinforcing the idea that peak flow is primarily influenced by precipitation inputs and catchment-wide hydrological responses rather than catchment or localized land cover patterns.

## 5. Discussion

### 5.1 The Role of Cultivation Practices on Peak Flow, Turbidity and Sediment Load at the hillslope-scale

Our results indicate that the effectiveness of non-erosive cultivation practices in reducing peak turbidity and sediment load in overland flow is highly dependent on field location relative to the stream and topography. In steep (~10–12% slope) and medium-to-distant agricultural areas (750–1000+ m from the stream), non-erosive practices did not significantly reduce sediment transport (Figure 6 and Figure 7; Table 3 and 4).

This aligns with previous studies (e.g., Boardman and Poesen, 2006), which emphasize that gravitational forces and high overland flow velocities in steep terrains enhance sediment detachment and transport. Even with increased vegetation cover, the steep topography and resulting high flow velocities in these areas likely promote continued sediment detachment, limiting the effectiveness of non-erosive cultivation in preventing soil erosion (Firoozi and Firoozi, 2024). Boardman and Poesen (2006) and Wu et al. (2018) demonstrated that sediment detachment intensifies in steep terrain due to accelerated overland flow, with runoff responses more dependent on infiltration capacity and surface roughness. This is supported by Zhang et al. (2020), who demonstrated that steeper slopes (>10%) experience enhanced soil loss, particularly when rainfall intensity is high. In flowpath 1, uninterrupted slopes (>10%) may accelerate overland flow, which may intensify sediment transport. As a result, non-erosive cultivation had minimal impact in these steep, medium-to-distant agricultural areas, as elevated flow velocities in these areas may sustain sediment detachment, thereby reducing the effectiveness of non-erosive cultivation in preventing soil erosion.

In contrast, non-erosive cultivation proved highly effective in flat agricultural areas close to the stream (7.2% slope, <500 m from the stream), significantly reducing turbidity (~3.8 times) and sediment load (~9.5 times). This aligns with findings from Lin et al. (2019), who demonstrated that winter wheat and cover crops can reduce sediment loss by up to 98.2% on moderate slopes, although their impact on total overland flow was limited. Similarly, Wu et al. (2019) found that wheat and other dense-cover crops effectively reduced sediment loss on moderate slopes (5–7%) but had limited ability to regulate total overland flow generation.

In Area C, vegetation effectively traps sediment before it reaches the E2 monitoring station. However, overland flow remains unchanged, as non-erosive practices primarily influence



sediment deposition rather than overland flow reduction. Here, vegetation density was
noticeably higher in non-erosive fields, as seen in field observations during event monitoring.
The denser root and canopy structure in non-erosive cultivation types like winter wheat
(Figure 10a) and cover crops (Figure 10b) compared to erosive cultivation types like maize
(Figure 10c and Figure 10d) slows sediment transport, allowing particles to settle because
overland flow slowed down due to the flatter terrain. Despite reducing turbidity and sediment
load, non-erosive practices had no impact on overland flow volume (Figure A1 and Table 3),
reinforcing that they regulate sediment transport rather than overland flow generation (Lin et
al., 2019).

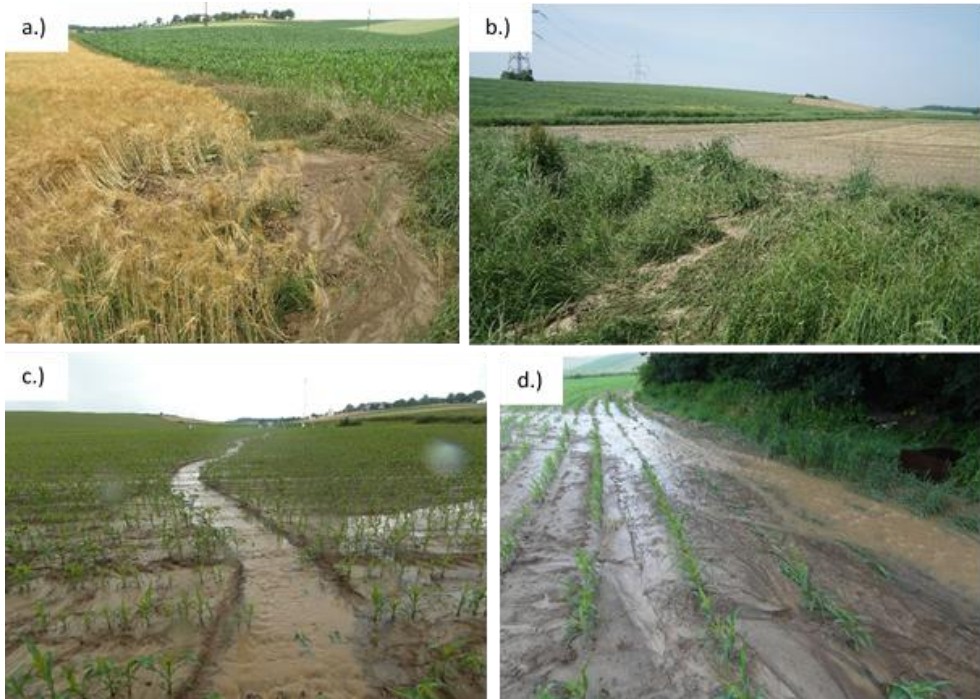


*Figure 10: The images at the top (a, b) illustrate how overland flow and sediment erosion originate from erosive areas of the*
*catchment, such as a) maize fields or b) bare soil patches, where limited vegetation cover fails to intercept overland flow. As*
*the sediment-laden water moves downslope, it reaches a field with non-erosive cultivation (a) winter wheat or b) cover crops),*
*where the dense vegetation effectively leads to sediment deposition, preventing further transport toward the stream. The*
*images at the bottom (c, d) illustrate overland flow and sediment erosion between the weather station and the stream. The*
*affected Area C is cropped with early-stage maize, which is an erosive crop species and does not effectively hinder overland*
*flow, leading to increased overland flow and sediment transport. (Photos: BAW-IKT)*
The influence of non-erosive cultivation on overland flow generation remained negligible
across all monitored locations (Figure A1). This supports findings from Wu et al. (2018) and
Zhang et al. (2020), which demonstrated that rainfall characteristics and soil moisture, rather
than land management, was the dominant control on overland flow generation in agricultural
catchments. Our findings indicate that non-erosive cultivation alone is insufficient to regulate
overland flow, in steep as well as in flat terrains. Given the high sediment transport in flowpath
1, additional conservation measures—such as contour plowing, buffer strips to reduce
sediment connectivity—are necessary to complement non-erosive practices. These strategies



would more effectively reduce sediment transport and enhance infiltration, particularly in
steeper agricultural areas (Borin et al., 2005).

## 5.2 The Role of Cultivation Practices on Peak Flow, Turbidity, and Sediment Load at the Catchment Outlet

Our results indicate that hydrological and sediment transport dynamics at the MW in-stream
monitoring site reflect contributions from multiple sources, including wetlands, springs,
subsurface flows, and direct overland flow from agricultural land. While the general
hydrological pattern observed at E2 persists, key differences emerge due to dilution and
additional flow contributions. Notably, in-stream sediment mobilization from riverbed and
riverbank erosion plays a negligible role in our catchment (Eder et al., 2014), suggesting that
the observed sediment loads are predominantly sourced from overland flow and subsurface
drainage pathways.
At the catchment outlet, peak flow increased between ~2.8 and ~11 times compared to E2,
indicating significant hydrological contributions from other flow contributions such as
wetlands, springs and tile drainages. This aligns with findings by Eder et al. (2014), who
demonstrated that subsurface flow and groundwater upwelling significantly contributed to
baseflow and peak discharge in the study catchment. The observed increase in peak flow
suggests that these additional flow contributions, which respond differently to precipitation
events, are critical in modulating stream discharge.
The magnitude of peak flow varied between erosive and non-erosive cultivation conditions at
Area C. While peak flow at E2 remained similar across both cultivation types, at MW, peak
flow was ~3 times higher under erosive compared to non-erosive conditions. Although non-
erosive cultivation may be generally associated with a ~20% reduction in overland flow (López-
Vicente and Navas, 2010), this trend was not observed at E2 in our study. Furthermore, the
observed increase in peak flow at MW cannot be fully explained by tile drainages, wetlands,
or springs, suggesting that unmonitored hydrological overland flow pathways play a larger role
than previously assumed.
Field observations during severe erosion events suggest that the E2 monitoring station
reached a capacity threshold, beyond which overland flow was no longer effectively captured.
Once this threshold was exceeded, a greater proportion of overland flow bypassed the
monitoring infrastructure through diffuse, unmonitored pathways, contributing directly to the
stream at MW. This interpretation aligns with hydrological connectivity and threshold
behavior studies (e.g., Saffarpur et al., 2016), which show that excess runoff can activate
alternative, less-monitored flow paths once critical flow thresholds are surpassed. As a result,
sediment and water inputs at MW during erosive events likely reflect both monitored
contributions from E2 and additional, unaccounted-for overland flow routed along these
alternative pathways.
At MW, peak turbidity decreased by ~5.4 to ~7.7 times relative to E2, indicating strong dilution
from wetlands, springs, and subsurface flow. This reduction exceeds typical values reported
for temperate agricultural watersheds (~3 to ~5 times, Sharpley et al., 2019), suggesting that
dilution processes in our catchment are particularly effective. Consistent with Vercruysse et



al. (2017), our findings confirm that groundwater, wetlands, and tributary inflows play a key role in regulating downstream sediment concentrations.

Despite dilution, the relative impact of cultivation at Area C on peak turbidity persisted from E2 to MW, emphasizing the dominant role of land management in controlling sediment transport (Boardman and Poesen, 2006). Specifically, erosive cultivation in Area C resulted in significantly higher peak turbidity at both E2 and MW, demonstrating that hillslope-scale sediment detachment has a persistent downstream impact (Zhang et al., 2020). In contrast, non-erosive cultivation in Area C effectively reduced peak turbidity at both monitoring sites, supporting findings from Lin et al. (2019) and Wu et al. (2019) that conservation practices (e.g., cover crops) reduce sediment detachment but do not necessarily alter total overland flow volumes.

While dilution lowered turbidity at MW, the relative difference between erosive and non-erosive cultivation became more pronounced. Under erosive cultivation, median peak turbidity at E2 was ~4 times higher than at MW, whereas under non-erosive conditions, it was ~8.5 times higher at E2 compared to MW. Furthermore, at E2, median peak turbidity was ~4.5 times higher for erosive compared to non-erosive conditions, while at MW, this difference increased to ~10.5 times.

A similar trend was observed for peak sediment load, where the effect of cultivation at Area C persisted at MW but was more pronounced. Under erosive cultivation, sediment load at MW was ~23.5 times higher than under non-erosive conditions, compared to ~10.5 times at E2. Additionally, peak sediment load at MW increased ~5.5 times relative to E2 under erosive conditions, whereas for non-erosive cultivation, the increase was only ~1.5 times.

These findings suggest that, similar to peak flow, peak sediment load under non-erosive conditions primarily originated from monitored overland flow at E2. However, during high-energy runoff events, when severe erosion occurred, additional unmonitored sources—such as tile drainage and diffuse overland flow—contributed significantly to sediment transport at MW. Field evidence indicates that once a critical connectivity threshold was exceeded, substantial overland flow bypassed the E2 monitoring point via alternative pathways that connected directly to the stream at MW, amplifying turbidity and sediment load. This observation aligns with threshold-driven hydrological connectivity in agricultural catchments, where high flow conditions activate previously disconnected pathways, allowing unmonitored sediment to reach downstream points like MW (Saffarpur et al., 2016).

Among unmonitored sediment sources, tile drainage systems—particularly Frau 1 and Frau 2—likely played a more significant role in sediment transport than previously assumed. These systems provide an alternative sediment pathway, bypassing overland flow routes and delivering fine sediments directly to the stream under specific hydrological conditions. Heavy precipitation events, high antecedent soil moisture, and preferential flow through macropores have been shown to trigger sediment-laden water transport via tile drains, circumventing traditional overland flow pathways (King et al., 2015).

Unlike overland flow, which primarily transports coarser sediment fractions, tile drains predominantly convey fine sediments, which can accumulate within the drainage pipes and



be intermittently flushed out during high-flow events. These sudden pulses of sediment
delivery are not necessarily linked to immediate overland flow, but rather occur when
hydrological connectivity between the fields and tile drains is established (Grangeon et al.,
2021; King et al., 2015). This intermittent flushing mechanism facilitates rapid and direct
sediment transfer to drainage outlets, thereby increasing sediment delivery to the stream
network (Grangeon et al., 2021). In some agricultural catchments, tile drains have been shown
to contribute to more than 50% of the annual sediment budget (Moore, 2016).
While tile drainage is unlikely to be the dominant sediment source in the HOAL catchment,
our findings suggest that it contributes more sediment than previously estimated. The tile
drains at Frau 1 and Frau 2 drain almost the entirety of Area C, representing a previously
unaccounted sediment transport pathway (Figure 3c,d). Additionally, the presence of mouse
holes in areas above the drainage system in Area C (Figure 11) suggests the formation of
preferential flow paths, which further enhance sediment transfer into tile drains. Such
preferential flow mechanisms accelerate sediment mobilization, allowing fine particles to
bypass natural retention and enter the stream unmonitored.

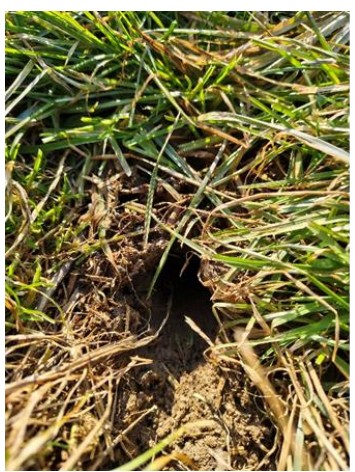
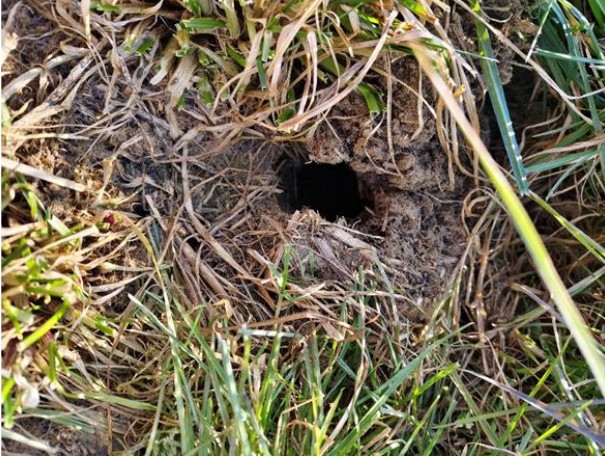


*Figure 11: Evidence of mouseholes above the tile drainage system in Area C. The images depict multiple small burrows created*
*by voles in grass-covered soil, forming macropores that enhance water and sediment infiltration. These macropores can serve*
*as preferential flow paths, facilitating rapid sediment transfer from agricultural topsoils into the tile drainage system,*
*particularly during high-flow events. This mechanism contributes to unmonitored sediment transport to drainage outlets,*
*reinforcing the role of subsurface connectivity in sediment mobilization. (Photos: BAW/Reinhard Hollerer)*

## 687    6. Conclusion

This study demonstrates that the effectiveness of non-erosive cultivation practices in reducing
turbidity and sediment load is highly dependent on topography, hydrological connectivity, and
field proximity to streams. Our findings at the hillslope scale (E2) indicate that non-erosive
cultivation significantly reduced peak turbidity (~9.5 times) and sediment load (~3.8 times) in
flat agricultural areas (<500 m from the stream, ~7.2% slope) but had no measurable effect in
steep (10–12% slope) or distant (>1000 m) agricultural fields, where high overland flow
velocities – driven by steeper topography – likely promote continued sediment detachment,
limiting the effectiveness of non-erosive cultivation in preventing soil erosion. Across all field
types, conversion to non-erosive cultivation had no significant effect on peak flow generation,



indicating that while these practices influence sediment transport, they do not regulate
overland flow volumes.
At the catchment scale (MW), dilution from subsurface flow reduced peak turbidity by 5.4–
7.7 times, whereas peak flow increased 2.8–11 times due to additional hydrological
contributions from wetlands, springs, and subsurface pathways. Peak sediment load at MW
was 2.4–5.4 times higher than at E2, suggesting that unmonitored diffuse overland flow and
tile drainage contribute to sediment mobilization at the catchment scale. These results
highlight the complexity of sediment connectivity in agricultural landscapes and the need for
multi-scale erosion control strategies.




Appendix: Additional figures


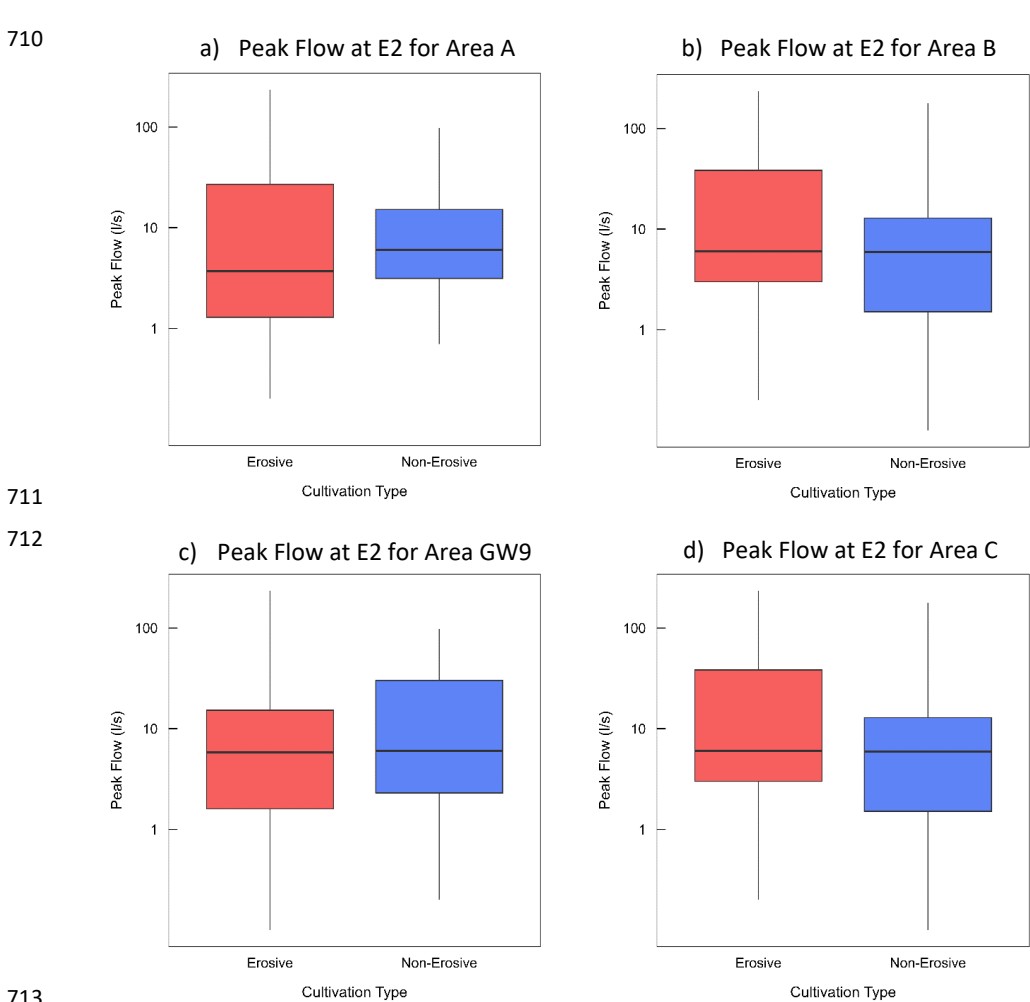


**Figure A1:** Boxplots showing the distribution of peak flow (l/s) for different cultivation types
(erosive vs. non-erosive) at overland flow measurement site E2 in Areas a) A, b) B, c) GW9,
and d) C. The median, interquartile range, and outliers are displayed for each category. The y-
axis is on a logarithmic scale.











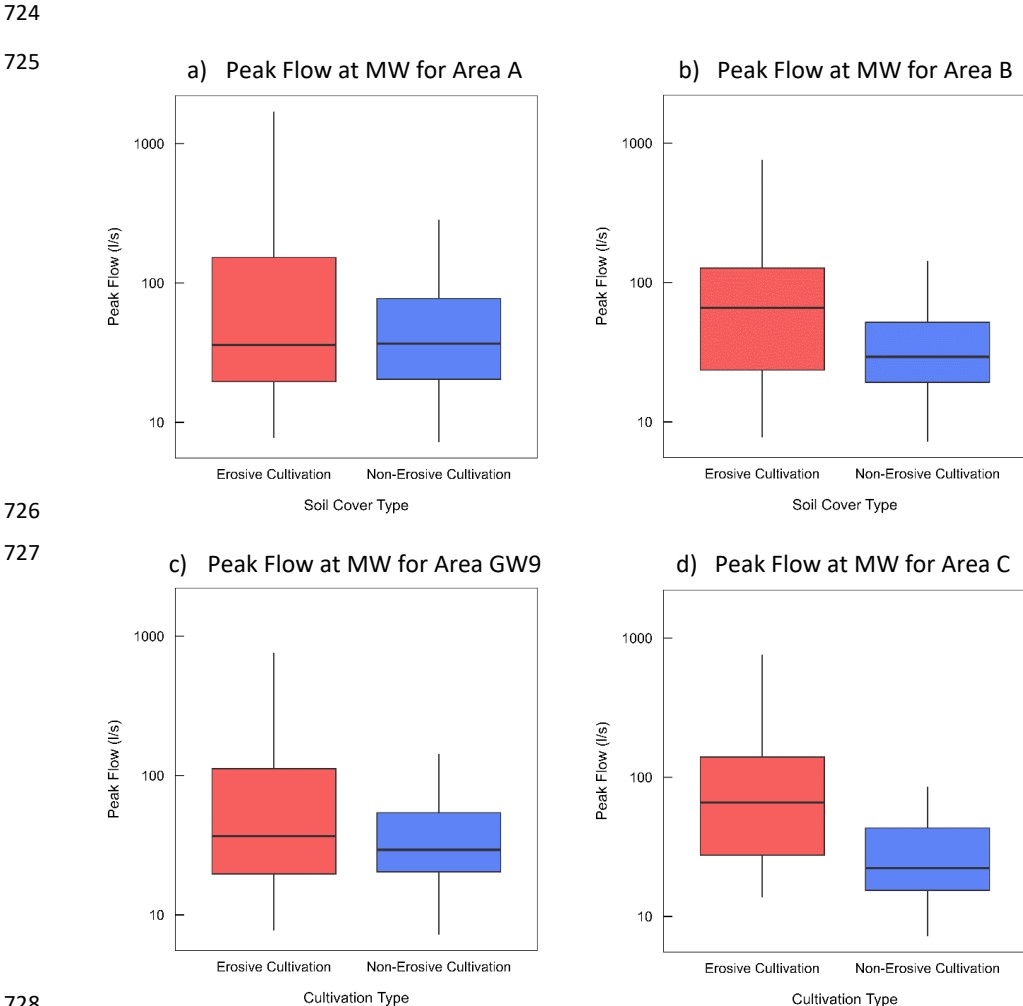




**Figure A2:** Boxplots showing the distribution of peak flow (l/s) for different cultivation types (erosive vs. non-erosive) at overland flow measurement site MW in Areas a) A, b) B, c) GW9, and d) C. The median, interquartile range, and outliers are displayed for each category. The y-axis is on a logarithmic scale.

733



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



## Code availability

The pieces of code that were used for all analyses are available from the authors upon request.

## Data availability

The datasets that have been analysed in this paper are available from the authors upon
request.

## Author contribution

CT: Conceptualization, Data Curation, Formal Analysis, Investigation, Methodology, Software,
Visualization, Writing (original draft preparation), Writing (review and editing)
BS: Data Curation, Conceptualization, Methodology, Writing (review and editing)
MB: Data Curation, Methodology, Writing (review and editing)
ES: Data Curation, Resources, Writing (review and editing)
CK: Data Curation, Resources, Writing (review and editing)
PS: Funding acquisition, Project administration, Supervision, Writing (review and editing)
GB: Funding acquisition, Project administration, Supervision, Writing (review and editing)

## Competing interests

At least one of the (co-)authors is a member of the editorial board of *Hydrology and Earth*
*System Sciences*.