# Peer review of "Identification of erosion hotspots and scale-dependent runoff controls on"

_EGUsphere, 2025_

## Author Comment (AC1)

**Identification of erosion hotspots and scale-dependent runoff controls on sediment transport in an agricultural catchment**

Christopher Thoma[1,2], Borbala Szeles[1,2], Miriam Bertola[1,2], Elmar Schmaltz[3], Carmen Krammer[3], Peter Strauss[3], Günter Blöschl[1,2]

[1] Institute of Hydraulic Engineering and Water Resources Management, Vienna University of Technology, Vienna, Austria

[2] Centre for Water Resource Systems, Vienna University of Technology, Vienna, Austria

[3] Institute for Land and Water Management Research, Federal Agency for Water Management, Petzenkirchen, Austria

**Corresponding author:** Christopher Thoma

**E-mail address:** thoma@hydro.tuwien.ac.at

30.07.2025

We sincerely thank the reviewer for their thoughtful and constructive feedback on our manuscript. We have fully addressed all the comments and propose revisions accordingly, as detailed below. Referee comments are shown in black, author replies are in blue. We believe these updates have resolved all the issues and look forward to further feedback from the editor and reviewer.

Kind regards,

Christopher Thoma and co-authors.

The manuscript presents an experimental analysis of the factor contributing to runoff, erosion and sediment load, at the small (66 ha) headwater catchment scale. The results suggest that considering both catchment structural connectivity and crop type (erosive vs non-erosive) is needed to assess the effect of management pratices on sediment load and peak flow.

The assessment of sediment source and field-to-stream connectivity at the catchment scale is a current research question. The additional effects of agricultural conservation practices on water and sediment dynamics at the catchment scale is an additional interesting and relevant scientific question. The studied catchment presents a high-quality database of traditional hydrological gauging stations, including high-frequency rainfall, runoff, streamflow and tile drainage monitoring of water and sediment load.

However, the manuscript presents major issues that preclude publication.

Below, we outline how we address all the issues raised by the reviewer.

1.) First, calculations are hard or not possible to understand. Particularly, the assessment of sediment load values, a central point in this study, is unclear. Was the turbidity-sediment concentration rating curve of good quality?

In our study, suspended sediment concentration (SSC) was derived by calibrating high-frequency turbidity measurements (FNU) against SSC values obtained from laboratory analyses of ISCO water samples collected during hydrological events. This calibration was performed separately for each station using paired turbidity–SSC data spanning a wide range of hydrological conditions. The turbidity–SSC relationship showed a strong and consistent fit across all events and sites ($R^2 = 0.86$ at site E2 and $R^2 = 0.98$ at MW).

Following this calibration, turbidity values were converted into SSC values in g/L, and the complete turbidity time series was thus expressed in sediment concentration units. Sediment loads were then calculated at 5-minute intervals using the formula:

$$\text{Sediment Load} \left(\frac{\text{g}}{\text{s}}\right) = \text{SSC} \left(\frac{\text{g}}{\text{L}}\right) \cdot Q \left(\frac{\text{L}}{\text{s}}\right)$$

These instantaneous loads were integrated over the duration of each event to derive total event-based sediment loads:

$$\text{Event Sediment Load} = \sum(\text{SSC} \cdot Q \cdot \Delta t)$$

where $\Delta t$ is the 5-minute time step.

We will clarify this methodology in the revised Methods section and include the turbidity–SSC calibration plots below for both stations in the Methodology or Appendix. We will also discuss the quality of the rating curves.

[Figure]

2.) The authors alternatively used turbidity and sediment load values in the analysis, but what is the point in analysing turbidity if sediment load values were available? Evaluating the robustness of the results is therefore not possible.

We agree that the use of both turbidity and sediment load values in the analysis may have introduced redundancy. We will remove all turbidity analyses.

3.) The methodology used for analysis is unclear. From my understanding, the authors chose to focus on peak values for flow and sediment/turbidity, which is surprising. To analyse the catchment dynamics, why not study the event-scale water volume and sediment load?

Yes, this is a very valid point. While, originally, our focus was on the peak values because of the larger number of sediment measurements around the peak, we have now changed the analysis to focus on the event-scale water volume and event-scale sediment load. In the revised paper we will back calculate the sediment concentrations for all time steps within the event based on turbidity-sediment concentration relationships and, where turbidity measurements are missing, based on discharge sediment concentration relationships from similar events or other time steps of the same event. This will allow us to estimate the complete event-based sediment loads and runoff volumes for all event.

4.) How did the authors account for hysteresis effects? How was the noise on turbidity values processed? Both may have significant implications for the robustness of the results, particularly considering the significant scattering presented in the log-log plots (Figure 4).

**Hysteresis effects:** The temporal dynamics of SSC, including any hysteresis effects, between discharge (Q) and SSC during hydrological events, are captured in our data because of the continuous turbidity measurement and numerous sediment samples available. The procedure will be described in more detail in the Methods section and we will also discuss the hysteresis effects in the revised paper.

**Processing noise on turbidity values:** Turbidity was measured using ViSolid700 IQ sensors installed at both locations (E2 and MW) in the HOAL catchment. These sensors operate on the nephelometric principle according to EN ISO 7027, using infrared light scattered at 90°, which

means that colour discoloration typically does not cause any interferences. Ultrasonic cleaning continuously ensures that the two glass windows – located at the bottom of the sensor – remain clean. This reduces long-term drift and noise contributions caused by contamination. The accuracy of the sensor is < 1% process variation coefficient. Repeatability is < 0.015%, i.e., there is very little measurement noise per measured value. In addition, the resolution automatically adjusts to the measuring range. At low turbidity values, ambient light or scattered light from walls can influence the measured value. For this reason, we must maintain a minimum distance of at least 10 cm from the water bottom or walls in order to minimize interference signals caused by reflections. For E2, this distance is approximately 17 cm, and for MW it is slightly more. The turbidity sensor is connected to a DIQ/S 182 WTW digital measuring and control device, which transmits the turbidity values directly to the logger. During data processing, the sensor data were aligned with laboratory reference values. These laboratory values are obtained either during events via automatic samplers or from weekly manual grab samples. Quantitative estimates of the measurement uncertainty will be included in the Methods section of the paper.

5.) The land use classification, which serves as a basis for analysis, is questionable. Defining winter wheat and winter barley as 'non-erosive' crops would require a strong justification, particularly in a study addressing the runoff event scale. What about the intra-annual variations of crops growing?

The classification of crops into erosive and non-erosive categories was based on the Austrian Agricultural Environmental Programme (ÖPUL) and the national Soil Erosion Evaluation Report (BML, 2021). These documents report that significantly lower average soil erosion rates were observed in regions with a higher proportion of non-erosive crops compared to erosive crops, particularly in our study region of Lower Austria. It is true that this binary classification may not fully capture the intra-annual variability in erosion risk at the event scale. The approach reflects a trade-off between classification granularity and the clarity and comparability of the analysis. To address this, we will clarify this limitation in the discussion section and emphasize that erosion risk may vary within crop types depending on phenological stage at the time of the event. We will also provide more detailed evidence on the erosion rates for the classification chosen.

6.) What about agricultural practices, e.g. storm event occurring on ploughed fields vs crusted fields? It is questionable to propose general results such as those proposed in this manuscript without combining the analysis of both soil surface and rainfall dynamics.

We did analyse the influence of agricultural practices, but we did not find any statistically significant effects on either runoff or sediment load. For this reason, the results were not included in the original manuscript. We will include the results of this analysis in the Appendix with a brief discussion in the main text.

7.) Moreover, it is unclear how the authors labelled the different areas (A, B, C, GW9) as 'erosive' or 'non-erosive' (e.g. in Table 2 and 4), considering that these areas included a mix of erosive and non-erosive crops (e.g. Area A in Figure 2). It is therefore not possible to assess if the main results are supported by the data.

Areas A and B consist of more than one field and thus occasionally contain a mix of erosive and non-erosive crops. The areas GW9 and C—which are central to the study's conclusions—comprise a single field with homogeneous cultivation classification during each event.

For analytical clarity and to enable statistically robust comparisons, we categorized areas A and B based on the dominant crop type at a given event: If >50% of the area was covered by non-erosive, the area was labelled as "non-erosive"; otherwise, if >50% of the area was covered by erosive crops it was considered as "erosive". This classification method was applied consistently across all events in the study period (2012–2022). We will clarify this classification approach in the Methods section. We will also acknowledge this limitation in the Discussion section of the revised manuscript, noting that this simplification may underestimate heterogeneity within areas, but was considered appropriate to facilitate spatially explicit, cross-scale analysis of erosion dynamics.

> 8.) Last, the main message of the manuscript, as indicated in the abstract and conclusion, i.e. the need to consider both cultivation practices and catchment connectivity, lacks novelty, particularly considering that only part of the catchment structural connectivity was considered in the analysis.

As highlighted by Vanmaercke et al. (2017), there remains a critical lack of long-term, high-frequency monitoring studies that can evaluate the effectiveness of erosion control measures across spatial scales. While conservation practices have been widely assessed at the hillslope scale, their effectiveness at the catchment scale remains poorly constrained—particularly regarding how overland flow and sediment redistribution shape in-stream sediment loads (Doody et al., 2017, Van Oost et al., 2007; Verstraeten et al., 2002). This gap limits our ability to assess conservation outcomes at early transport stages, as long-term studies linking surface runoff contributions to catchment-scale sediment connectivity remain scarce despite advances in erosion monitoring. Our study directly addresses this gap by using a high-resolution, multi-station dataset from the HOAL catchment, including rainfall, streamflow, tile drainage, and sediment monitoring at event scale over an entire decade. We explicitly account for area-specific characteristics such as slope, proximity to the stream network, catchment connectivity and catchment area specific cultivation and tillage practices. This allows us to move beyond previous work by empirically quantifying both sediment source dynamics and field-to-stream connectivity at the catchment scale—addressing a key research need for multi-scale sediment transport assessments identified by Vanmaercke et al. (2017). We have now sharpened the conclusions to better bring out the novelty of the research.

> 9.) As a conclusion, given the lack of novelty, the unclear analysis procedure, and the inability to assess whether the results support the claims presented, I would not recommend publication in HESS.

We have carefully addressed all general and specific comments and have clearly outlined how we will revise the manuscript to improve clarity, strengthen the analysis, and better highlight the novel contributions of our study.

**Specific comments**

10.) The title is misleading: 'identifying erosion hotspots' would require dedicated studies using e.g. sediment tracing and/or distributed modelling.

We agree that the term "identifying erosion hotspots" may imply the use of sediment fingerprinting or distributed modelling approaches, which were not applied in this study. To avoid any misunderstanding regarding the methods used, we will revise the title of our study, also considering the views of the other reviewers.

11.) Figure 2 is hardly readable, please consider bringing monitoring stations to the foreground and / or to increase dots size. It may help readers to use intuitive names for the monitoring stations over the manuscript, i.e. what is the difference between 'Sys' and 'Frau'? Would it be relevant to change these names for TD (Tile Drain) and other monitoring stations for e.g. S (Streamflow), R (Runoff)… ?

We will increase the dot size of Figure 2, enhance the contrast, and bring the monitoring stations to the foreground to improve readability.

Regarding the station naming, we understand the reviewer's concern. However, we would retain the original station abbreviations (e.g., "Sys4", "Frau2") because they are consistent with long-term monitoring at the HOAL and are widely used in previous publications from this catchment. To improve clarity for readers unfamiliar with these abbreviations, we will add a short explanation of each station's dominant flow pathway (e.g., tile-drainage, streamflow) to the figure caption and re-emphasize this in the Methods section.

12.) It is misleading to provides Pearson's r on a scatterplot including regression lines. It is suggested to add correlation coefficients to the correlation matrices, and to indicate determination coefficients in the figures.

Since we now include total event runoff volume and total event sediment load directly in the revised manuscript, this plot will no longer be part of the revised manuscript.

13.) Table 3: It is surprising that the peak flow is not significantly affected by tile drainage. Literature underlines the importance of tile drainage in modulating peak flow.

We completely agree with reviewer that tile-drainages play a key role in modulating peak flow. In fact, we were highlighting the modulation of peak flow by tile-drainages in lines 94-96 in our Introduction section, and in lines 596—602 in our Discussion section. We also discussed the dilution of peak turbidity at the catchment-scale due to tile drainage contributions compared to the hillslope-scale.

14.) l.493-497: It is not clear how a correlation coefficient can be used to deduce that 'rainfall erosivity exerts a dominant control over hydrological and sediment transport mechanisms'. It is also surprising that the correlation between EI30 and the sediment dynamics is better at the catchment scale than at the plot scale, while increasing scale should results in a higher complexity.

We agree that correlation alone does not prove causality or "dominant control" in a mechanistic sense. We will revise the sentence in line 493–497 to better reflect the descriptive, rather than causal, nature of the observed relationship.

15.) It is also surprising that the correlation between EI30 and the sediment dynamics is better at the catchment scale than at the plot scale, while increasing scale should results in a higher complexity.

We thank the reviewer for this insightful observation. Indeed, it may seem counterintuitive at first that the $EI_{30}$–sediment dynamics relationship improves with increasing scale, given the expectation of added complexity. However, as discussed in Sivapalan (2003), increasing spatial scale can lead to emergent simplicity. That is, aggregation of hillslope processes can filter out localized variability and highlight dominant controls (such as rainfall erosivity or connectivity patterns), which results in stronger, more stable correlations at the catchment scale–as also observed in our analysis. We will add this perspective and cite Sivapalan (2003) in the revised manuscript to clarify this point.

**References**

Doody, D. G., Archbold, M., Foy, R. H., and Flynn, R.: Approaches to the implementation of the Water Framework Directive: Targeting mitigation measures at critical source areas of diffuse phosphorus in Irish catchments, Journal of Environmental Management, 93, 225–234, https://doi.org/10.1016/j.jenvman.2011.09.002, 2012.

Sivapalan, M.: Process complexity at hillslope scale, process simplicity at the watershed scale: is there a connection?, Hydrological Processes, 17, 1037–1041, https://doi.org/10.1002/hyp.5109, 2003.

Van Oost, K., Quine, T. A., Govers, G., De Gryze, S., Six, J., Harden, J. W., Ritchie, J. C., McCarty, G. W., Heckrath, G., Kosmas, C., Giraldez, J. V., da Silva, J. R. M., and Merckx, R.: The impact of agricultural soil erosion on the global carbon cycle, Science, 318, 626–629, https://doi.org/10.1126/science.1145724, 2007.

Vercruysse, K., Grabowski, R. C., and Rickson, R. J.: Suspended sediment transport dynamics in rivers: Multi-scale drivers of temporal variation, Earth-Science Reviews, 166, 38–52, https://doi.org/10.1016/j.earscirev.2016.12.016, 2017.

Verstraeten, G., Van Oost, K., Van Rompaey, A., Poesen, J., and Govers, G.: Evaluating an integrated approach to catchment management to reduce soil loss and sediment pollution through modelling, Soil Use and Management, 18, 386–394, https://doi.org/10.1111/j.1475-2743.2002.tb00257.x, 2002.

---

## Author Comment (AC2)

Identification of erosion hotspots and scale-dependent runoff controls on sediment transport in an agricultural catchment

Christopher Thoma1,2, Borbala Szeles1,2, Miriam Bertola1,2, Elmar Schmaltz3, Carmen Krammer3, Peter Strauss3, Günter Blöschl1,2

1 Institute of Hydraulic Engineering and Water Resources Management, Vienna University of Technology, Vienna, Austria

2 Centre for Water Resource Systems, Vienna University of Technology, Vienna, Austria

3 Institute for Land and Water Management Research, Federal Agency for Water Management, Petzenkirchen, Austria

**Corresponding author:** Christopher Thoma

E-mail address: thoma@hydro.tuwien.ac.at

17.10.2025

We sincerely thank you for your thoughtful and constructive assessment of our manuscript. We greatly appreciate your positive comments regarding the relevance of our study, the quality of the writing, and the clarity of our figures.

We have carefully addressed all the major and minor comments and propose revisions accordingly, as detailed below. Referee comments are shown in black, author replies are in blue. We believe these updates have strengthened the manuscript and clarified the points raised.

Kind regards,

Christopher Thoma and co-authors.

The text is sound and can be accepted for publication at HESS. It tackles an interesting and relevant topic (erosion hotspots and their relation to runoff and sediment transport) and uses primary data for the analysis. the text is well written and the figures are meaningful. Despite its merits, the manuscript needs some improvement before publication.

Below, we outline how we address all the comments raised by the reviewer.

**Major review.**

1) Lines 117-119. You mention "sediment load change across spatial scales" as one of your aims. However, the experimental catchment and the stream reach are small (66 ha and 620 m, respectively). Do you consider that this experimental setup allows you to investigate different "spatial scales"?

Thank you for this valuable comment. We agree that the overall spatial extent of the HOAL catchment (66 ha) is small compared to large river basin studies. Our intention with the term "spatial scales" was not to imply regional or basin-scale variability, but rather to analyse sediment transport dynamics across different hydrological source areas within the catchment — specifically, (i) the hillslope-scale at E2 (45 ha), and (ii) the catchment-scale at MW (66 ha). We will clarify this by explicitly referring to "hillslope-scale versus catchment-scale" instead of the more general term "spatial scales" and revise our research question accordingly. This distinction is still meaningful as flow pathways differ substantially between these monitored locations. E2 only consists of overland flow at the hillslope-scale, whereas the catchment outlet MW aggregates multiple flow pathways such as perennial and intermittent spring discharges, perennial and intermittent tile-drainage systems, and wetlands as described in lines 148-175 of the manuscript.

2) 3. Methods. A table with the data availability (e.g., number of samples, monitoring period, method...) would be desirable. Besides, neither a table, nor a figure with the events' characteristics is presented, which hampers the readership possibility of analyzing the results.

Thank you very much for this helpful suggestion. In the revised manuscript, we will add a table summarizing data availability (monitoring period, number of samples, and measurement methods for discharge and turbidity). In addition, we will include a separate overview table of all analyzed events, including start and end time, event duration, rainfall amount, Qmax, total flow volume, and total sediment load.

3) Line 234. How did you measure the total kinetic energy?

Thank you very much for pointing this out. We will clarify this in the revised manuscript. The total kinetic energy E of rainfall was calculated following McGregor et al. (1995) as shown in equation (1):

$$E = 0.29 \cdot [1 - 0.72^{(-0.082 \cdot I_{30})}]$$
 (1)

where  $I_{30}$  is the maximum 30-minute rainfall intensity during the event (mm h-1), and E is the kinetic energy of that rainfall event expressed in MJ ha-1 mm-1. The resulting  $EI_{30}$  estimates were comparable to previously reported results for the HOAL (e.g., Szeles et al., 2025).

4) Lines 271-272. Please justify the selection of the trigger values (5 Ls-1, 2 Ls-1, 100 mgL-1) and comment how these values may intefere on the results.

Thank you for pointing this out. The trigger values ( $5 L s^{-1}$ ,  $2 L s^{-1}$ ,  $100 mg L^{-1}$ ) were originally adopted following Eder *et al.* (2010) to minimize the inclusion of low flow events with negligible sediment transport. We would like to clarify that only events that triggered overland flow at E2 were ultimately included in the analysis (n = 55). Events with values close to these thresholds generally do not generate overland flow and therefore do not contribute to the dataset used for comparisons between E2 and MW. Consequently, while the precise thresholds influence the total number of events identified at the catchment outlet MW (n = 255), they do not affect the subset of events representing overland flow at E2, which were used for our analysis. These threshold values thus do not interfere with our results. We agree that listing these trigger values in the manuscript is not essential and will remove them to avoid confusion.

5) Figures 8 and 9. Why should erosive land cover (especially at this spatial scale) correlate with EI30? Please consider revising the figures.

Thank you for this insightful comment. We agree that the correlation between  $El_{30}$  and the percentage of erosive land cover involves two conceptually independent variables. In the revised manuscript, we will remove this correlation to avoid implying a causal relationship.

6) 6. Conclusion. We expect to read some conclusive statements about hydrological and sedimentological processes (e.g., dillution, sediment sources, and macropores, among others).

Thank you very much for this constructive suggestion. We agree that the current conclusion can better emphasize the underlying hydrological and sedimentological processes. In the revised manuscript, we will extend the conclusion to explicitly discuss key mechanisms, including (i) dilution at the catchment outlet, where subsurface inflows from springs, wetlands, and tile drains increase flow volume but contribute little sediment, resulting in lower SSC compared to the hillslope outlet E2, and (ii) episodic sediment transfer through preferential flow pathways such as macropores and vole burrows above the tile drainages, which we documented in the field (Figure 11), facilitating rapid surface-to-subsurface connectivity and allowing fine sediment to bypass retention. The updated conclusion will therefore provide a clearer process-based synthesis beyond the summary of observed results.

**Minor review.**

7) Lines 56-57. I understand that the numbers (20%, 80%) refer to a specific example, they cannot be used in generalized terms. Please consider rephrasing the sentence.

We agree that the original phrasing may imply a generalization. We will rephrase the sentence to clarify.

8) Figure 2. Please identify the height of the isolines.

We will label the heights of the isolines in Figure 2 directly on the contour lines.

9) Lines 287-290. Please check for the correct reference to Figures 4a, 4b, 4c, and 4d.

Since we now include total event runoff volume and total event sediment load directly in the revised manuscript, this plot will no longer be part of the revised manuscript.

10) Figures 4b and 4d. Flow volumes vary two orders of magnitude for similar peak flow. How valid is the use of peak flo as a proxy in these cases?

Yes, this is an important point and was also raised by Reviewer 1. It has now been fully addressed in the revised manuscript.

While, originally, our focus was on the peak values because of the larger number of sediment measurements around the peak, we have now changed the analysis to focus on the event-scale water volume and event-scale sediment load. In the revised paper we will back calculate the sediment concentrations for all time steps within the event based on turbidity-sediment concentration relationships and, where turbidity measurements are missing, based on discharge sediment concentration relationships from similar events or other time steps of the same event. This will allow us to estimate the complete event-based sediment loads and runoff volumes for all event.

11) Table 2. Please provide the size of each area. This is particularly relevant to compare the results at E2 and MW.

In the revised manuscript, we will add the size of each area in Table 2. Additionally, we will add to the Study Area description of the revised manuscript that the catchment size for the E2 station is 45 ha and was derived from a high-resolution 1 m LiDAR DEM using flow accumulation and watershed extraction in ArcGIS Pro. This allowed precise delineation of hydrologically nested contributing areas from the hillslope-scale E2 (45 ha) to the catchment-scale at MW (66 ha).

12) The term EI30 is (correctly) defined as 'erosivity', but throughout the text and figure captions, we read that EI30 is "intensity". Please revise it.

We thank the reviewer for pointing this out. We will carefully revise the manuscript and all figure captions to ensure that  $El_{30}$  is consistently referred to as 'erosivity' instead of 'intensity.'

13) Lines 657-658. Where were the data from Frau 1 and Frau 2 presented? What do you mean by "previously assumed"?

In the revised manuscript, we will present the data for the tile-drainages Frau2 and Frau1 and replace the term "previously assumed" with a more accurate description. Specifically, we intended to convey that the contribution of the tile-drainages is likely higher than the measured values suggest, because some sediment is deposited in the metal H-flume before reaching the turbidity sensor (Figure 4d in the manuscript).

**References**

Eder, A., Strauss, P., Krueger, T., & Quinton, J. (2010). Comparative calculation of suspended sediment loads with respect to hysteresis effects (in the Petzenkirchen catchment, Austria). *Journal of Hydrology, 389*, 168–176. https://doi.org/10.1016/j.jhydrol.2010.05.043.

McGregor, K. C., Bingner, R. L., Bowie, A. J., & Foster, G. R. (1995). Erosivity index values for northern Mississippi. *Transactions of the American Society of Agricultural Engineers, 38*(4), 1039–1047. https://doi.org/10.13031/2013.27950.

Szeles, B., Parajka, J., Šraj, M., Blöschl, G., Marjanović, D., Bezak, N., Lebar, K., Vidmar, A., Strauss, P., Krammer, C., Schmaltz, E., Hogan, P., Rab, G., & Zabret, K. (2025). Comparative analysis of rainfall event characteristics and rainfall erosivity between two experimental plots in Austria and Slovenia. *Journal of Hydrology: Regional Studies, 59*, 102353. https://doi.org/10.1016/j.ejrh.2025.102353.

---

## Author Comment (AC3)

**Identification of erosion hotspots and scale-dependent runoff controls on sediment transport in an agricultural catchment**

Christopher Thoma[1,2], Borbala Szeles[1,2], Miriam Bertola[1,2], Elmar Schmaltz[3], Carmen Krammer[3], Günter Blöschl[1,2], Peter Strauss[3]

[1] Institute of Hydraulic Engineering and Water Resources Management, Vienna University of Technology, Vienna, Austria

[2] Centre for Water Resource Systems, Vienna University of Technology, Vienna, Austria

[3] Institute for Land and Water Management Research, Federal Agency for Water Management, Petzenkirchen, Austria

**Corresponding author:** Christopher Thoma

**E-mail address:** thoma@hydro.tuwien.ac.at

06.01.2025

We sincerely thank Prof. John Quinton for the time he took to provide his detailed, thoughtful and constructive feedback, which significantly helped us to improve the quality of the manuscript. We have addressed all the comments and propose revisions accordingly, as detailed below. Prof. Quinton's comments are shown in black, author replies are in blue. We hope these updates have resolved all the issues and look forward to further feedback.

Kind regards,

Christopher Thoma and co-authors.

1.) This is an interesting paper that provides a lot of detail on the transfer of sediment through the HOAL catchment. However, because of the volume of information the paper struggles to move from a detailed study of the HOAL catchment to a paper which highlights findings that have wider interest to the hydrological community beyond those working on HOAL. This is a major weakness and will require a major revision. Currently the paper reads like a report or chapter on the HOAL catchment rather than a paper suitable for publication in HESS.

We thank the reviewer for this insightful comment and agree. We have therefore restructured the narrative to highlight transferable processes, mechanisms, and event-scale sediment-transport dynamics relevant to agricultural catchments beyond the HOAL. We have reduced the descriptive content and worked on the cross-station synthesis so that the focus lies on hydrological processes rather than site-specific characteristics. We clarified the broader relevance of the HOAL: Its diverse hydrological and sediment transport pathways (tile drainages, overland flow, wetlands, springs) provide a natural laboratory in which different sediment flow pathways and transport processes can be studied. This diversity makes HOAL representative of a wider range of agricultural catchments, as also emphasized by Blöschl *et al.* (2016). We hope, that the revised manuscript now more clearly articulates how the findings relate to agricultural catchments in the alpine foreland, central Europe, and other regions with comparable soil types, precipitation regimes, land-uses, and cultivation practices.

2.) The analysis is largely based around means and deviations around the means for different characteristics of the hydro/sedigraphs. Not surprisingly there is a lot of variability which makes it hard to see if there are any differences. I wonder if there are better ways of analysing these time series which pair the data in some way. For example paired ratios or erosive to non-erosive land use for individual events. This requires a significant effort.

Thank you for this helpful and constructive comment. We have now added information on the event-based differences between erosive and non-erosive cultivation by analysing the ratio of erosive to non-erosive conditions in relation to event size (=$EI_{30}$).

For Areas A, B, and GW9, event sizes are evenly distributed between erosive and non-erosive cultivation. This indicates that the observed differences are not an artefact of event-size.

For Area C, however, event sizes are unevenly distributed: larger events predominantly occurred during erosive cultivation, whereas smaller events occurred during non-erosive cultivation. To assess whether the previously identified significant differences are biased by this uneven distribution, we re-analysed discharge, suspended sediment concentration, and sediment load for Area C while explicitly accounting for event-size.

At the hillslope-scale (E2), the results remain unchanged, and the significant effect of cultivation in Area C persists even when controlling for event size. In contrast, at the catchment-scale (MW), the results change: when accounting for event-size, cultivation in Area C no longer shows a significant effect on suspended sediment concentration or sediment load.

To improve visual interpretability, we revised the existing boxplots by adding individual event points, with point size representing $EI_{30}$.

I have made a large number of comments below.

Below, we outline how we address all the comments raised by Prof. John Quinton.

3.) L142  Figure 2.

3.1) It is hard to see these points on the map. Perhaps include an inset map with the stream

We agree. We have increased the dot size of Figure 2, enhanced the contrast, and brought the monitoring stations to the foreground to improve readability.

3.2) Is the non erosve cultivation always in the same place?

Non-erosive cultivation occurs throughout the catchment, and its location and extent varies from year to year. In addition to Table 1 and Figure 2, which shows the spatial distribution of erosive and non-erosive cultivation for the year 2015 as an example, we have prepared maps showing the annual distribution of erosive and non-erosive cultivation for each year of the study period. These are provided in the Supplementary Material.

3.3) Needs to separate out the pathways from the monitoring points. For example: 'this pathway is important for these reasons and is monitored at this point using this kit'

Thank you for this comment. We fully agree and have revised the manuscript to clearly separate the description of the hydrological and sediment transport pathway from the description of the monitoring points.

4.) L149 Figs 3a and b are flumes not gullies. E1 and E2 are presumably flumes. Separate the monitoring from the features

We fully agree and have revised the manuscript so that E1 and E2 are now described as flumes used to monitor overland flow pathways.

5.) L154 Same point. these are flumes not tile drainage systems.

We fully agree and have revised the manuscript so that Sys1-Sys4 and Frau1-Frau2 are now described as flumes used to monitor tile-drainage pathways.

6.) L156 Is this from the surface or subsurface?

Thank you for highlighting this ambiguity. The exact origin of the sediment transported by tile drainage systems cannot be uniquely attributed to either surface or subsurface sources. Sediment delivery via tile drains likely represents a combination of surface-derived material entering the drainage network through preferential flow pathways (e.g. small burrows created by voles that form macropores) and subsurface-derived fine sediment originating from the soil matrix or the drainage infrastructure itself. We have revised the manuscript to explicitly state this mixed and uncertain sediment origin and to clarify the associated transport mechanisms.

Frau 1 and Frau 2 were not monitored due to financial constraints. Available project resources were prioritised towards a limited number of stations to ensure continuous long-term data collection.

7.) L180 Again separate monitoring from the form/pathway. e.g. a) Flume used to monitor an overland flow pathway

Thank you for this comment. We fully agree and have revised the manuscript to clearly separate the description of the hydrological and sediment transport pathway from the description of the monitoring points.

8.) L188 Is this a combination/rotation of crops or is only one crop grown per year ? Not clear

The arable land is managed as a rotation of different crop types; each field is planted with only one crop per year. Occasionally, additional cover crops are used after the harvest of the main crop, but never in combination with maize, only grain crops. Across the catchment, multiple crop types are present at any given time. A map with a typical cultivation for the arable land in the year 2015 is presented in Figure 2 of the manuscript. We have clarified this in the revised manuscript and additionally provide a file in the Supplementary Material showing the planting and harvesting schedule for each field.

Yes, the grassland is permanent. We have clarified this in the revised manuscript.

Formulation was changed.

11.) L207 How good was this relationship? Was it affected by particle size? You need to tell us.

In our study, suspended sediment concentration (SSC) was derived by calibrating high-frequency turbidity measurements (FNU) against SSC values obtained from laboratory analyses of ISCO water samples collected during hydrological events. This calibration was performed separately for each station using paired turbidity–SSC data spanning a wide range of hydrological conditions. The turbidity–SSC relationship showed a strong and consistent fit across all events and sites ($R^2$ = 0.86 at site E2 and $R^2$ = 0.98 at MW). Particle size did not systematically affect the relationship, as the turbidity sensors responded consistently.

We have clarified this in the revised Methods section and included the turbidity–SSC calibration plots below for both stations in the Methodology or Appendix. We will also discuss the quality of the rating curves.

[Figure]

[Figure]

12.) L221 It would be good to have the planting and harvesting schedule as supplementary information. Could be a data base file

Yes, we agree and will provide a file in the Supplementary Material showing the planting and harvesting schedule for each field in the catchment.

13.) L249 Table 1. Can you give us a spatial feel for where the erosive cropping takes place? I realise that it might be impractical to have ten maps in the main paper, but perhaps in the suplementary information/

Erosive cultivation occurs throughout the catchment, and its location and extent vary from year to year. In addition to Table 1 and Figure 2, which shows the spatial distribution of erosive and non-erosive cultivation for the year 2015 as an example, we have prepared maps showing the annual distribution of erosive and non-erosive cultivation for each year of the study period, which are provided in the Supplementary Material.

14.) L254 define direct flow. Is it the same as overland flow? make sure you are consistent with your terms

Yes, "direct flow" refers to overland flow in this context. We have revised the manuscript to use the term overland flow consistently throughout.

15.) L317 Replace This with Thus, the …

The text has been corrected.

16.) L320 can you provide a distance in m?Long and shorter could mean 1 cm or 1 km

Are these slope are so different? The range of slopes appears b to be quite narrow (9.7 to 11.5%) - I certainly wouldn't describe 11.5% as very steep. I suggest you just refer to the slope steepnesses in %

Thank you for this comment. We fully agree and have revised the entire manuscript to report slope steepness in percent and distances in meters, rather than using subjective terms.

**17.) L351 Avoid these subjective steepness terms**

We fully agree and have revised the entire manuscript to report slope steepness in percent instead of using subjective terms.

**18.) L354 You have this information in the text. Either have it in a table or text, but not both**

Thank you for this comment. We have retained the table and revised the text to refer to the table.

**19.) L392. Avoid subjective descriptors of steepness and distance**

Same reply as before: We fully agree and have revised the entire manuscript to report slope steepness in percent and distances in meters, rather than using subjective terms.

**20.) L401 quote to p<0.01**

The text has been corrected.

**21.) L404 How did you get g/l as your turbidity unit. Did you calibrate and if so how good was the calibration and did you take account of that uncertainty when testing for differences. This needs to be described**

In our study, suspended sediment concentration (SSC) was derived by calibrating high-frequency turbidity measurements (FNU) against SSC values obtained from laboratory analyses of ISCO water samples collected during hydrological events. This calibration was performed separately for each station using paired turbidity-SSC data spanning a wide range of hydrological conditions. The turbidity-SSC relationship showed a strong and consistent fit across all events and sites ($R^2 = 0.86$ at site E2 and $R^2 = 0.98$ at MW).

Following this calibration, turbidity values were converted into SSC values in g/L, and the complete turbidity time series was thus expressed in sediment concentration units.

We clarified this methodology in the revised Methods section and included the turbidity-SSC calibration plots for both stations. The quality of the calibration was very high, and the associated uncertainty is small relative to the observed variations in SSC; therefore, it does not materially affect the statistical analyses of differences between events.

**22.) L407 Figure 6. I wonder if there is a better way of looking at this data since you clearly have a lot of variability caused by different event sizes which makes it hard to see differences. Could you for example look at the ratios of Erosive:Non Erosive and see how that relates to event size?**

**Otherwise you have lot of graphs which aren't very interesting and could probably be dropped from the manuscript**

Thank you very much for this very helpful and constructive comment. We have now investigated the influence of event-size on the observed differences between erosive and nonerosive cultivation by analysing the ratio of erosive to non-erosive conditions in relation to event size ($=EI_{30}$).

For Areas A, B, and GW9, event sizes are evenly distributed between erosive and non-erosive cultivation. This indicates that the observed differences are not an artefact of event-size. To improve visual interpretability, we have revised the boxplots by adding points for each individual event, with point size representing $EI_{30}$.

For Area C, however, event-sizes are unevenly distributed: larger events predominantly occurred during erosive cultivation, whereas smaller events occurred during non-erosive cultivation. To assess whether the previously identified significant differences are biased by this uneven distribution, we re-analysed discharge, suspended sediment concentration, and sediment load for Area C while explicitly accounting for event-size.

At the hillslope-scale (E2), the results remain unchanged, and the significant effect of cultivation in Area C persists. In contrast, at the catchment-scale (MW), the results change: when accounting for event size, cultivation in Area C no longer shows a significant effect on suspended sediment concentration or sediment load.

23.) L416 But this is data from across all events where the variability will prevent you finding significant differences. The data needs to be paired in some way - see my previous comment.

Thank you very much for this clarification. As detailed in our response to Comment 22, we addressed this issue by explicitly accounting for event-size ($=EI_{30}$). All results for Areas A, B, and GW9 remain non-significant, while cultivation effects for Area C remain significant at the hillslope-scale (E2), even when accounting for event-size.

24.) L444. I am not convice by the graphs in Figure 7. They seem to repeat the information in Table 4.

We have revised Figure 7 and now retain only the plot for Area C, where statistically significant differences were identified, including the revised visualisation accounting for event-size as described above. All plots for non-significant results (Areas A, B, and GW9) have been removed, such that Figure 7 now provides visual support for the statistical results summarized in Table 4.

25.) L471 Figure 8 looks like a Table to me. Which location does the analysis relate to? Some of the correlations reported are non sensical. Why for example would you expect a correlation between EI30 and erosive area? Do not present correlations for things which you know are not correlated. Comment also applies to 'Figure 9'

The analysis in Figure 8 relates to the hillslope-scale station E2, as indicated by the sub-chapter heading "4.2.1 Overland Flow Characteristics," while Figure 9 relates to the catchment-scale station MW, as indicated by "4.2.2 In-stream Measurement Characteristics." We have renamed these headings to "Hillslope-scale (E2)" and "Catchment-scale (MW)" for consistency and clarity, and also specified this in the figure captions.

We fully agree that the correlation between $EI_{30}$ and the percentage of erosive land cover involves two conceptually independent variables, as also noted in the community comment (CC1) by Prof. José Carlos de Araújo. In the revised manuscript, we have removed this correlation to avoid implying a causal relationship.

26.) L550  The areas you refer to are not flat!  They have a slope of 7.2%!

Same reply as before: We fully agree and have revised the entire manuscript to report slope steepness in percent, rather than using subjective terms.

27.) L560 Here and in other places in the discussion (Figure 10, Figure 11) you introduce new results in the form of observations. These need to be in the results.

Thank you very much for this comment. We agree that new quantitative results or novel analytical findings regarding the spatial-scale effects should be presented exclusively in the results section. We have therefore revised the manuscript and moved the quantitative findings of the spatial-scale analysis from the discussion to the results section.

The descriptions associated with Figures 10 and 11, however, to us do not introduce new measured results, but rather provide qualitative, illustrative field observations intended to support the quantitative results presented earlier in the results section. Figures 10 and 11 are therefore used as visual example of erosion and deposition processes (e.g., erosive vs. non-erosive cultivation) that were already quantified and presented based on monitoring data in the results section. Thus, we suggest to keep these figures in the discussions section.

We have revised the discussion section accordingly to explicitly frame these descriptions as illustrative field evidence supporting the results, and to avoid wording that could be interpreted as introducing new results. We also clarified references to the corresponding quantitative findings.

28.) L580 This is an important finding. Put it at the front of the paragraph then discuss it.

Thank you for pointing this out. We agree that this is a key finding and have revised the paragraph so that it now appears at the beginning and is discussed immediately thereafter.

29.) L627 This seems like a an important finding which has more generic value than some of the very site specific findings that have been discussed above.  I would recommend making it more prominent.

Thank you for this valuable comment. We agree that this finding has more general applicability beyond the HOAL-specific results. We have therefore made it more prominent in the discussion by clarifying its broader implications.

30.) L637-646 Reads like results rather than discussion

Thank you for pointing this out. We have moved this information to the results section.

31.) L671 A key point that is worthy of discussion. Place it at the top of the paragraph then discuss.

Thank you for this suggestion. We agree that this is a key point and have moved it to the beginning of the paragraph, followed by the corresponding discussion.